# HALF-ORDER FINE-TUNING FOR DIFFUSION MODEL: A RECURSIVE LIKELIHOOD RATIO OPTIMIZER

**Tao Ren**[1][*][†] **Zishi Zhang**[1][*] **Jinyang Jiang**[1][*] **Zehao Li**[1][*] **Shentao Qin**[2,3][*] **Yi Zheng**[1]
**Guanghao Li**[2] **Qianyou Sun**[1] **Yan Li**[4] **Jiafeng Liang**[5][‡] **Xinping Li**[1] **Yijie Peng**[1,6][‡]
[1] Peking University   [2] Tsinghua University   [3] OriginFlow   [4] HKUST
[5] Harbin Institute of Technology   [6] Xiangjiang Laboratory

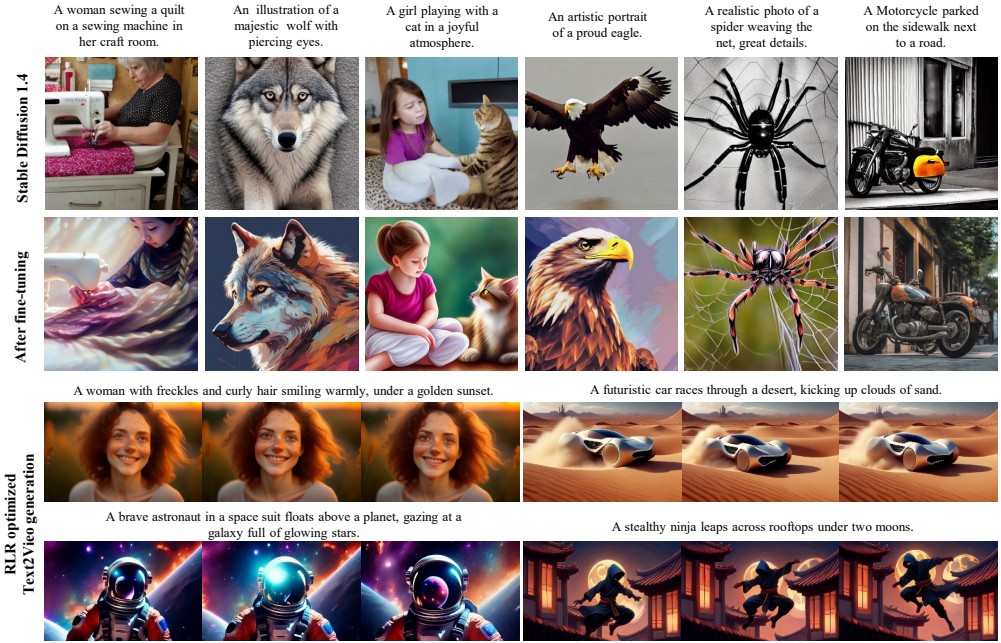

Figure 1: Image and video samples generated from the model fine-tuned by our RLR optimizer. Please refer to Appendices D, E, and F for more qualitative examples.

## ABSTRACT

The probabilistic diffusion model (DM), generating content by inferencing through a recursive chain structure, has emerged as a powerful framework for visual generation. After pre-training on enormous data, the model needs to be properly aligned to meet requirements for downstream applications. How to efficiently align the foundation DM is a crucial task. Contemporary methods are either based on Reinforcement Learning (RL) or truncated Backpropagation (BP). However, RL and truncated BP suffer from low sample efficiency and biased gradient estimation, respectively, resulting in limited improvement or, even worse, complete training failure. To overcome the challenges, we propose the Recursive Likelihood Ratio (RLR) optimizer, a Half-Order (HO) fine-tuning paradigm for DM. The HO gradient estimator enables the computation graph rearrangement within the recursive diffusive chain, making the RLR's gradient estimator *an unbiased one with lower variance* than other methods. We characterize the bias, variance, and convergence behavior of our method. Extensive experiments are conducted on image and video generation to validate the superiority of the RLR. Furthermore, we propose a novel prompting technique that is natural for the RLR to achieve a synergistic effect. The implementation is available at `https://github.com/RTkenny/RLR-Optimizer`.

---

[*] Equal contribution.   [†] Project lead: `rtkenny@stu.pku.edu.cn`
[‡] Corresponding author: `jfliang@ir.hit.edu.cn, pengyijie@pku.edu.cn`

# 1 INTRODUCTION

Probabilistic diffusion model (DM) (Sohl-Dickstein et al., 2015; Ho et al., 2020; Zhang et al., 2024b; Gao & Li, 2025) has emerged as a transformative framework in high-fidelity data generation, demonstrating unmatched capabilities in diverse applications such as image synthesis (Podell et al., 2023), video generation (Wang et al., 2023a), and multi-modal data modeling. These models operate by recursively denoising latent representations, as shown in Figure 2, effectively capturing complex data distributions. However, fine-tuning DMs in the post-training phase remains a daunting challenge, since the gradient estimation through the recursive structure imposes excessive computation overhead (Clark et al., 2023). This challenge has limited the broader deployment of DM in dynamic and resource-constrained environments.

It is a natural way to fine-tune DMs via full backpropagation (BP) through all time steps (Rumelhart et al., 1986), which is theoretically functional, providing precise gradient estimation over the entire diffusion chain. However, the computational and memory overhead of BP scales prohibitively with the model size and the number of diffusion steps (Prabhudesai et al., 2023; Yuan et al., 2024a; Clark et al., 2023; Prabhudesai et al., 2024), making full BP impractical for most real-world scenarios. Specifically, training Stable Diffusion 1.4 by full BP with a batch size of 1 and 50 time steps would require approximately 1TB of GPU RAM (Prabhudesai et al., 2023). Thus, truncating recursive differentiation becomes a common practice to alleviate memory overhead. But truncated BP suffers from structural bias, as it terminates gradient computation before sourcing to the input, only considering a limited subset of the diffusion chain. Fine-tuning based on a biased gradient estimation can inadvertently impair the optimization performance, resulting in **model collapse**, i.e., contents generated reducing to pure noise. Empirical evidence, in Figure 3, shows that truncated gradients result in a significant drop in reward scores during training: the fewer the truncated time steps, the more severe the model collapse. Moreover, truncated BP **fails to capture the multi-scale information** across all time steps due to the absence of differentiation on early steps. The hierarchical feature of generation imposes a thorough gradient evaluation to retain fidelity from pixel to structural level.

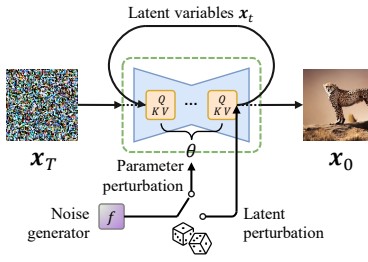

Figure 2: The recursive structure of diffusion models for gradient estimation.

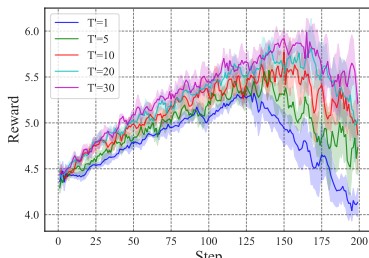

Figure 3: Model collapse caused by the truncation: training SD 1.4 on the aesthetic reward model by truncated BP.

Reinforcement learning (RL) (Schulman et al., 2017) as a gradient computation trick has enabled another branch of DM fine-tuning (Lee et al., 2023; Black et al., 2023; Wallace et al., 2024; Fan et al., 2024). It typically ignores the differentiable connection between steps and recovers the gradient by estimation. RL avoids caching intermediate activations, significantly reducing memory requirements. It also supports gradient computation in a divided manner under extreme circumstances to accommodate insufficient memory. The cost to pay is the high variance of the estimated gradient. Even if the estimator is unbiased, the variance can result in wild sample-inefficient updates, demonstrated by the slow convergence during training.

These limitations of BP and RL underscore the necessity of a more efficient, scalable, and stable fine-tuning approach that harmonizes computational tractability with optimization efficacy. To address this, we first investigate the recursive architecture of the DM and propose the problem of finding the minimal variance gradient estimator. Informed by perturbation-based estimation using Likelihood Ratio (LR) techniques (Jiang et al., 2024; Ren et al., 2025) (detailed reviews of LR techniques are provided in the Appendix A), we propose the Recursive Likelihood Ratio (RLR) optimizer. By utilizing the inherent noise in the DM, a local computational graph is enabled for pathwise gradient estimation as shown in Figure 4, which can reduce the variance and better capture the multi-scale information. The RLR estimator is **unbiased and has lower variance under the same computation budgets** as other methods. Through perturbation-based computational graph rearrangement, the RLR mitigates the structural bias of truncated BP and the high variance of RL, capable of better capturing multi-scale visual information. Our optimizer shares similarity with zeroth-order optimizer since both

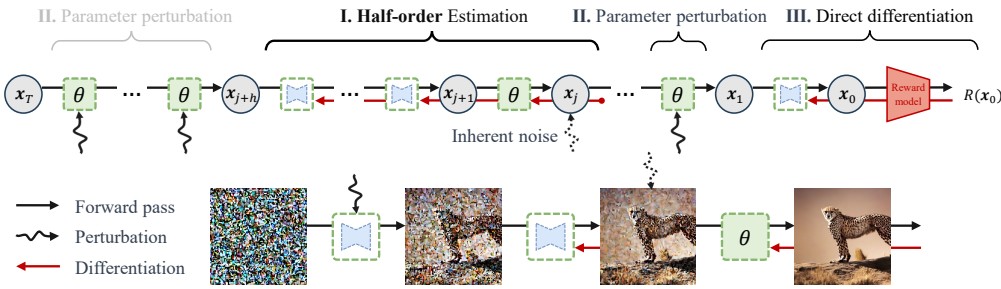

Figure 4: The computation paradigm of the RLR optimizer.

need perturbation to estimation gradient, but the RLR utilize a local BP chain to enable low-variance estimation. Therefore, we name it as **Half-Order** optimizer. Our contributions are threefold:

- We provide a systematic analysis of gradient estimation in DMs, identifying a structured design space of estimators. Then, we formulate the RLR optimizer in the design space under the protocol of minimizing variance with a limited computational budget.

- Extensive evaluation of Text2Image and Text2Video tasks are conducted. RLR consistently achieves higher reward scores across multiple human preference reward models and outperforms SOTA video models on the VBench benchmark. Furthermore, we propose a prompt technique for our RLR, validating the intuition and applicability of our method.

- We conduct a rigorous theoretical analysis of RLR's design, proving its unbiasedness, bounding its estimator variance, and establishing convergence guarantees. These results formally justify the empirical success of RLR and explain the deficiency of prior methods.

## 2 RELATED WORK

Diffusion probabilistic models have achieved state-of-the-art performance in multi-modal generation (Ho et al., 2020; Rombach et al., 2022). However, aligning pre-trained DMs for downstream tasks remains challenging. RL and supervised post-training have been widely explored to incorporate human preferences or task-specific objectives, with recent methods including RLHF-inspired approaches, DPO variants, and reward-model-guided fine-tuning (Ziegler et al., 2019; Fan et al., 2024; Wallace et al., 2024; Prabhudesai et al., 2023). Yet, RL-based fine-tuning often suffers from high variance and sample inefficiency, while truncated BP reduces memory cost but introduces structural bias that can lead to model collapse(Prabhudesai et al., 2023; 2024; Xu et al., 2024b). Recent forward-learning methods based on stochastic gradient estimation offer a promising alternative with lower computational and memory costs (Salimans et al., 2017; Peng et al., 2022; Chen et al., 2023). Please refer to Appendix A for extended related works.

## 3 UNBIASED MINIMAL VARIANCE GRADIENT ESTIMATOR FOR DIFFUSION MODEL

### 3.1 PROBLEM FORMULATION

DMs generate data by transforming a noise sample $x_T$ into a data sample $x_0$ through a recursive denoising process, as shown in Figure 2. This generation proceeds through $T$ steps of stochastic updates, each governed by a parameterized mapping $\phi_t$, where the implementation of $\phi_t$ may vary across steps, yielding the transformation:

$$x_{t-1} = \phi_t(x_t, z_t; \theta) \tag{1}$$

where $\theta$ denotes the model parameter, and $z_t = \sigma_t \epsilon_t, \epsilon_t \sim \mathcal{N}(\mathbf{0}, I)$ is a stochastic perturbation. The noise term $z_t$ can be either the inherent noise in the DM, i.e., $x_{t-1} = \varphi(x_t; \theta) + z_t$, or the injected noise to the parameter, i.e., $x_{t-1} = \varphi(x_t; \theta + z_t)$. The function $\varphi$ denotes the shared backbone network of the DM, reused across different steps, while $\phi_t$ should be used with the subscript $t$ to indicate which time-step it operates on. The full generation chain can be represented recursively as:

$$x_0 = \phi_{1:T}(x_T, z_{1:T}; \theta) = \phi_1 \circ \phi_2 \circ \cdots \circ \phi_T(x_T, z_{1:T}; \theta) =$$
$$\phi_1(\phi_{2:T}(x_T, z_{2:T}; \theta), z_1; \theta) = \phi_1(\phi_{2:T-1}(\phi_T(x_T, z_T; \theta), z_{2:T-1}; \theta), z_1; \theta), \tag{2}$$

where $z_{1:T}$ are independent noise terms injected at each step. Once a pre-trained model is available, a reward model $R(x_0)$ is introduced to guide post-training towards generating samples of higher semantic or perceptual quality. The resulting fine-tuning objective becomes:

$$\max_\theta \mathbb{E}[R(x_0)] = \max_\theta \mathbb{E}_{z_{1:T}}[R(\phi_{1:T}(x_T, z_{1:T}; \theta))]. \tag{3}$$

Fine-tuning DMs through the recursive structure poses a critical challenge: how to efficiently and accurately estimate the gradient of the expected reward with respect to model parameters. A natural objective is to construct a *gradient estimator* for $\mathbb{E}[R(x_0)]$ that is **unbiased** and has **minimal variance**, under a given computational and memory budget. We formulate the problem as the following constrained optimization problem over the space of gradient estimators:

$$\min_{G \in \mathcal{G}} \quad \text{Var}(G) \qquad \text{s.t.} \quad \nabla_\theta \mathbb{E}[R(x_0)] = \mathbb{E}[G], \quad \mathcal{C}(G) \leq \mathcal{B}, \tag{4}$$

where $\mathcal{G}$ denotes the space of all *unbiased* gradient estimators $G$ for the objective $\mathbb{E}[R(x_0)]$, with the sample $x_0$ being obtained via a generative chain defined by $x_0 = \phi_{1:T}(x_T; z_{1:T}, \theta)$. The term $\text{Var}(G)$ refers to the variance of the estimator, which we aim to minimize. The cost function $\mathcal{C}(G)$ quantifies the total computational and memory overhead incurred by the estimator $G$, typically measured in terms of the length of backward passes or the volume of intermediate activations stored. Finally, $\mathcal{B}$ represents the computational budget available for gradient estimation.

## 3.2 $\mathcal{G}$: The Unbiased Gradient Estimator Design Space in Diffusion Models

We now characterize the feasible design space $\mathcal{G}$ for unbiased gradient estimators in DMs. Due to the recursive structure of the generative chain, the gradient estimator must propagate through all $T$ steps. At each step $t$, we may choose one of three gradient estimation strategies:

- **First-Order (FO)** uses exact backpropagation through $\phi_t$. **Zeroth-Order (ZO)** perturbs the parameters directly, e.g., $\varphi_t(x_t; \theta + \sigma_t \epsilon_t)$, and estimates the gradient via $\frac{R(\varphi(\cdot; \theta + \sigma_t \epsilon_t))}{\sigma_t} \epsilon_t$, based solely on function evaluations (Salimans et al., 2017).

- We propose unbiased **Half-Order (HO)** gradient estimator which utilizes the inherent noise $z_t$ (rather than extrinsic perturbation) and applies the Likelihood Ratio technique, producing an estimator of the form: $R(x_0) \cdot D_\theta^\top \phi_{t:t+h-1} \cdot \nabla \log f(z_t)$, where $D_\theta \phi_{t:t+h-1}$ denotes the Jacobian of a local sub-chain of length $h$, and $f(\cdot)$ denotes the noise density. The HO allows a $h$-length sub-chain, starting from any $t$ in the diffusive chain. RL is a special case of HO method with $h = 1$.

The full estimator design space is thus defined as:

$$\mathcal{G}_{\text{full}} := \{G = (g_1, \ldots, g_T) \mid g_t \in \{\text{FO}, \text{HO}, \text{ZO}\} \ \forall 1 \leq t \leq T\}, \tag{5}$$

where each sequence $G \in \mathcal{G}_{\text{full}}$ represents a composite estimator composed of local choices at each time step. Table 1 summarizes the variance, unbiasedness, and memory costs of each strategy: **FO has the lowest variance but highest cost, ZO is the cheapest but suffers from the highest variance, and HO balances the two.** The theoretical underpinnings of these variances, unbiasedness, and memory comparisons are formalized in Section 6, and constitute an independent contribution of this work.

Table 1: Comparison of gradient estimators.

| Method | Unbiasedness | Variance | Memory |
|---|---|---|---|
| First-order | ✓ | Small | Large |
| Half-order | ✓ | Medium | Medium |
| Zeroth-order | ✓ | Large | Small |
| Truncated BP | ✗ | Small | Medium |

# 4 Solving the Variance Minimization Problem: Recursive Likelihood Ratio Optimizer

## 4.1 Recursive Likelihood Ratio Optimizer

Having characterized the decision space $\mathcal{G}_{\text{full}}$, we now turn to solving the variance minimization problem (4). Due to practical constraints, we further reduce the full design space by enforcing the following structure: (i) All HO estimators should be connected in the chain, since

separating the HO path would incur high variance. (ii) FO should be directly connected to the reward model, according to its definition. This way, the decision space $\mathcal{G}_{\text{full}}$ is reduce to $\mathcal{G}_{\text{RLR}} = \{(g_1^{\text{FO}}, \cdots, g_j^{\text{HO}}, \cdots, g_{j+h}^{\text{HO}}, \cdots, g_T^{\text{ZO}}) \mid 1 \le j \le T - h\}$, which consists of one FO at the first step, a HO sub-chain of length $h$ starting at step $j$, and ZO estimators for all remaining steps to ensure unbiasedness. Under this specific structure, the decision variables are reduced to the length $h$ and the starting index $j$. Each solution in $\mathcal{G}_{\text{RLR}}$ is referred to as a Recursive Likelihood Ratio (RLR) estimator (see Figure 4), which takes the form

$$
\begin{aligned}
G = \quad &\underbrace{D_\theta^\top \phi(x_1, z_1; \theta) \frac{dR(x_0)}{dx_0}}_{\text{One-step first-order estimator}} \\
&- \underbrace{R(x_0) D_\theta^\top \phi_{j:j+h}(x_{j+h}, z_{j:j+h}; \theta) \nabla_z \ln f(z_j)}_{h-\text{length half-order estimator}} - \underbrace{\sum_{i \in C} R(x_0) \nabla_z \ln f(z_i)}_{\text{zeroth-order estimator}},
\end{aligned}
\tag{6}
$$

where $j \in [1, T - h] \cap \mathbb{Z}$, $C = \{1, 2, \ldots, T\} \setminus \{j, j+1, \ldots, j+h\}$, $z_j \sim \mathcal{N}(\mathbf{0}, \sigma_j I)$, and $z_i \sim \mathcal{N}(\mathbf{0}, \sigma_i I)$ for $i \in C$.

**Differentiating the reward model.** In the first part of the RLR estimator, we apply the FO estimator to the first time step to directly backpropagate through the reward model, avoiding black-box treatment (e.g., RL and ZO) and better leveraging its structure, as shown in Figure 4.

**Fixed-horizon half-order optimization.** The generation process of the DM follows a coarse-to-fine structure, with every time step in the chain controlling a different scale of generation. Incorporating precise gradient information from every time step is essential, but full BP is computationally prohibitive. Truncated BP introduces bias, while ZO and RL lead to high variance by ignoring structural information. To address this, the RLR optimizer incorporates an HO $h$-length sub-chain, capturing multi-scale information while minimizing variance. Specifically, the starting index of HO, $j \sim \mathcal{J}(1, T - h)$, is randomly selected across the whole diffusive chain, following a given distribution $\mathcal{J}$ (see Section 3.2). The inherent perturbation, $z_j$, enables the localized $h$-length sub-chain, $D_\theta \phi_{t:t+h-1}$, effectively capturing the visual scale information represented around that step (see Section 3.2 for choosing $h$).

**Surrogate estimator via parameter perturbation.** For the remaining times steps, $C = \{1, 2, \ldots, T\} \setminus \{j, j+1, \ldots, j+h\}$, we inject noise directly into the model's parameters to construct ZO estimation to ensure unbiasedness. This approach is computationally cheap without caching intermediate latent variables.

## 4.2 Optimizing $h$ and $j$: Variance–Memory Tradeoffs

The remaining task is to optimize the two variables in the RLR estimator, $h$ and $j$, to solve the optimization problem (4). Notably, the choice of $j$ does not directly affect this surrogate objective, but instead influences the ability to capture multi-scale information across different steps, so we treat its decision as a separate problem of interest.

**Optimizing $h$.** To reduce the number of problem parameters that need to be estimated, we use an upper bound on the variance of the RLR estimator (see the Appendix J.4) as a surrogate objective:

$$
\min_{h \in \mathbb{N}_0: \ G(h) \in \mathcal{G}_{\text{RLR}}} \sum_{t=1}^T \text{Var}(g_t) + 2 \sum_{t \ne t'} \sqrt{\text{Var}(g_t) \text{Var}(g_{t'})}
\tag{7}
$$
$$
\text{s.t.} \quad \mathcal{B}_h h + \mathcal{B}_z (T - 1 - h) \le \mathcal{B},
$$

where $h$ and $(T - 1 - h)$ are the number of steps for HO and ZO; $\mathcal{B}_h$ and $\mathcal{B}_z$ are coefficients indicating the magnitude of the memory cost of HO and ZO per step. In practice, the available budget satisfies $\mathcal{B}_z T < \mathcal{B} < \mathcal{B}_h T$, meaning that using pure ZO underutilizes the budget, while using pure HO exceeds it. Let $V_h^2$ and $V_z^2$ denote the per-step variance of the HO and ZO, respectively. Since HO

and FO have much lower variance than ZO, we use a common $V_h$ for both HO and FO, and assume $V_h \ll V_z$ and $T > 2$. These conditions ensure the optimization problem admits the solution:

$$h^* = \min\{\lfloor \frac{\mathcal{B} - \mathcal{B}_z(T-1)}{\mathcal{B}_h - \mathcal{B}_z} \rfloor, \lfloor \frac{TV_z}{2(V_z - V_h)} - 1 \rfloor\} > 0. \tag{8}$$

We set $\mathcal{B}_h = 8\text{GB}$ and $\mathcal{B}_z = 0.24\text{GB}$, which is supported by empirical evidence in Table 9 in the Appendix. If the memory budget $\mathcal{B}$ is between 30GB and 40GB. It is recommended to set $h = 2$. As the formula (7) indicates, the variance decreases as $h$ increases. However, the performance exhibits diminishing improvement with increasing $h$. In other aspects, the memory consumption grows linearly with $h$, and the computational time grows even more rapidly. The above claims are corroborated by our ablation results in Table 9. Therefore, even with a larger memory budget, blindly increasing $h$ is not advisable. Moreover, since $V_h \ll V_z$, the second term in (8) simplifies to approximately $\frac{T}{2} - 1 \approx 24$, which is typically larger than the first term. As a result, we have $h^* = \lfloor (\mathcal{B} - \mathcal{B}_z(T-1))/\mathcal{B}_h - \mathcal{B}_z \rfloor$ in practice, and there is no need to estimate $V_h$ and $V_z$.

**Determining $j$.** We use the gradient norm to represent the importance of different steps. Then sample $j$ from the categorical distribution $j \sim \mathcal{J}(1, T-h) = \mathcal{CAT}(\text{Softmax}(\|g_1\|, \cdots, \|g_{T-h}\|))$.

## 5 EXPERIMENTS

We verify the superiority and applicability of the RLR optimizer against various baselines on two generation tasks: Text2Image and Text2Video. We compare the RLR with the RL-based method (DDPO), and the truncated-BP-based methods (Alignprop and VADER). Moreover, other baselines, e.g, closed-source models, are also included. Finally, we propose a novel prompt technique that is natural for the RLR optimizer, demonstrating the enhanced capability of the proposed RLR optimizer to capture multi-scale information for visual generation. Ablations are included to verify the validity of the proposed RLR optimizer. Please refer to Appendix B for detailed settings.

Table 2: Text to Image reward score. We evaluate methods under different DM under different reward models. The higher the score, the better the performance.

| Model | Methods | Pick-a-Pic | | | | HPD v2 | | | |
|---|---|---|---|---|---|---|---|---|---|
| | | PickScore | HPSv2 | AES | ImageReward | PickScore | HPSv2 | AES | ImageReward |
| SD1.4 | Base | 16.24 | 21.03 | 4.48 | 32.74 | 16.19 | 22.08 | 4.42 | 32.90 |
| | DDPO | 17.56 | 23.15 | 5.47 | 49.33 | 17.53 | 22.79 | 5.52 | 52.06 |
| | Alignprop | 18.08 | 26.64 | 5.91 | 65.07 | 19.17 | 27.02 | 6.02 | 67.18 |
| | **RLR** | **20.14** | **28.57** | **6.53** | **75.65** | **21.38** | **29.22** | **6.65** | **76.55** |
| SD2.1 | Base | 16.37 | 22.14 | 4.53 | 35.40 | 16.25 | 23.32 | 4.57 | 36.03 |
| | DDPO | 17.70 | 24.55 | 5.58 | 52.48 | 17.43 | 24.56 | 5.62 | 52.85 |
| | Alignprop | 19.23 | 27.20 | 6.07 | 68.09 | 21.60 | 27.40 | 6.11 | 72.62 |
| | **RLR** | **22.58** | **30.11** | **6.66** | **77.26** | **23.22** | **30.98** | **6.74** | **83.07** |

Table 3: Text2Video Generation Evaluation on the Vbench. The weighted average is calculated by assigning a weight of 1 to all metrics, except for the Dynamic Degree metric, which is assigned a weight of 0.5.

| Methods | Subject Consistency | Background Consistency | Motion Smoothness | Dynamic Degree | Aesthetic Quality | Imaging Quality | Weighted Average |
|---|---|---|---|---|---|---|---|
| VideoCrafter | 95.44 | 96.52 | 96.88 | 53.46 | 57.52 | 66.77 | 79.97 |
| Pika | 96.76 | **98.95** | 99.51 | 37.22 | 63.15 | 62.33 | 79.87 |
| Gen-2 | 97.61 | 97.61 | **99.58** | 18.89 | **66.96** | 67.42 | 79.75 |
| T2V-Turbo | 96.28 | 97.02 | 97.34 | 49.17 | 63.04 | **72.49** | 81.96 |
| DOODL | 95.47 | 96.57 | 96.84 | 55.46 | 58.27 | 66.79 | 80.30 |
| DDPO | 95.53 | 96.63 | 96.92 | 58.29 | 59.23 | 66.84 | 80.78 |
| VADER | 95.79 | 96.71 | 97.06 | 66.94 | 66.04 | 69.93 | 83.45 |
| **RLR** | **97.64** | 97.19 | 98.05 | **70.69** | 66.15 | 71.08 | **84.63** |

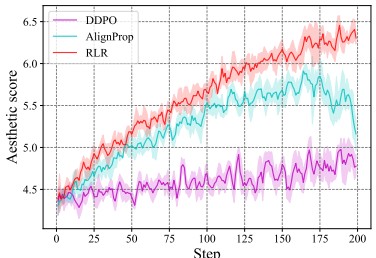
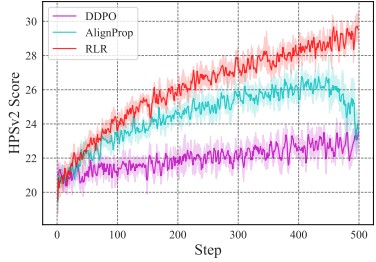

(a) Reward curves for aesthetic score.  (b) Reward curves for HPS v2 score.

Figure 5: Sample efficiency analysis, reward curves under the same training steps for SD 1.4.

## 5.1 TEXT2IMAGE GENERATION

We evaluate our methods on two DMs: Stable Diffusion 1.4 and 2.1 (Rombach et al., 2022). As shown in Table 2, the RLR methods achieve higher reward scores on the unseen prompts from the test set. The RL-based method have limited improvement with respect to the base model, due to the sample inefficiency nature. Alignprop has considerable improvement over the base model. However, the biased estimator limits its further improvement. Training details and hyperparameters can be found in the appendix.

**Sample efficiency analysis.** The compare the sample efficiency and the variance of different methods, we show the reward curves of SD 1.4 when training on the AES and HPS v2 models in Figure 5. The DDPO optimizes the reward at a very slow pace, indicating high variance and low sample efficiency. In the earlier phase, the AlignProp has comparable performance as the RLR. In the later phase, while the RLR can continue to improve the reward, the AlignProp suffers from severe model collapse.

## 5.2 TEXT2VIDEO GENERATION

We compare our RLR not only with RL and truncated BP but also with a series of open-source or API-based Text2Video models. In the metric of DD and AQ, the RLR surpasses other methods by a large margin. In other metrics, RLR achieves considerable improvement over the base model, VideoCrafter. Some API-based models have better performance on some metrics, but the gaps are small. In terms of the weighted average score, our RLR has the best performance over all baselines.

## 5.3 DIFFUSIVE CHAIN-OF-THOUGHT

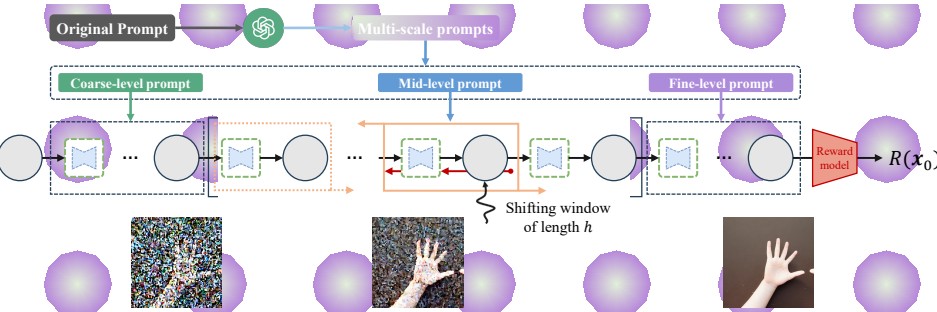

Figure 6: The framework of Diffusive Chain-of-Thought. The DM generates images in a multi-scale manner: earlier steps for low-resolution features and later steps for high-resolution features. If a specific scale has deficiencies, we utilize the HO estimator to enhance the corresponding steps.

Furthermore, we propose the Diffusive Chain-of-Thought (DCoT), a prompt technique that is natural for our RLR optimizer to demonstrate the applicability of our method. The core idea is that **DMs generate content in a multi-scale (coarse-to-fine) manner**, and deficiencies at a particular scale can be addressed by **focusing gradient updates on the corresponding steps** of the diffusion process by the HO sub-chain. We propose dividing all the diffusion process steps into three groups: coarse-level, mid-level, and fine-level. The coarse-level chain includes steps adjacent to the initial noise, focusing

on generating a rough outline. The fine-level chain includes steps adjacent to the final output, focusing on the fine-grained details. The mid-level chain in between focuses on the geometric structure of the content.

The idea of DCoT is shown in Figure 9, which converts the original prompt into multi-scale prompts reflecting the coarse-to-fine nature. Different generation steps should be conditioned on different prompts instead of being conditioned on the same prompt. The HO estimator term in the RLR enables a $h$-step local computational chain for low-variance, unbiased gradient esti-

Table 4: Experiment results for Diffusive Chain-of-Thought

| Methods | PickScore | HPSv2 | AES | ImageReward |
|---|---|---|---|---|
| SD1.4 | 15.33 | 20.50 | 4.76 | 33.45 |
| SD1.4-DCoT | 16.38 | 21.68 | 4.83 | 39.24 |
| RLR w/o | 18.05 | 23.58 | 5.27 | 43.09 |
| **RLR-DCoT** | **19.45** | **25.80** | **5.83** | **49.88** |

mation. By integrating DCoT, we can target the HO sub-chain precisely at the time steps (i.e., scales) where generation is deficient, as revealed by the multi-scale prompt decomposition. The HO estimator term uniformly picks a starting point $j \sim \mathcal{U}(1, T - h)$ from the entire $T$-step chain to start the local $h$-step BP chain. When applying the DCoT to the fine-tuning process, we should constrain the sample range of $j$ in the steps where deficiencies exist, $j \sim \mathcal{U}(a, b), 1 < a < b < T - h, b - a > h$. In our experiment for the hand task, we set $a = 30$ and $b = 40$.

We write 5 prompts for hand generation and then prompt ChatGPT to generate the multi-scale prompts for the three levels. We report the performance in Table 4. As shown in the table, either simply applying DCoT to the base model or combining it with the RLR can improve the performance significantly. The RLR with the DCoT has the best performance. Qualitative results are shown in the Appendix F.

## 5.4 ABLATION STUDY

We conduct the ablation study, using SD 1.4 and HPD v2, to verify the contribution of different parts in the RLR optimizer. In Table 5, we evaluate the RLR and its variants (V1: the RLR without HO and ZO; V2: the RLR without ZO; V3: the RLR without HO). The V1 performs the worst since

Table 5: Ablation of the RLR.

| Methods | PickScore | HPSv2 | AES | ImageReward |
|---|---|---|---|---|
| RLR w/o HO & ZO | 18.43 | 23.66 | 5.78 | 60.07 |
| RLR w/o ZO | 20.11 | 27.07 | 6.23 | 68.35 |
| RLR w/o HO | 19.28 | 26.70 | 5.92 | 63.85 |
| RLR | 21.38 | 29.22 | 6.65 | 76.55 |

it actually reduces to the truncated BP with only one time-step. The V2 and V3 perform better than the V1. It is worth noting that the V2 is better than the V3. The V3 without HO is actually an unbiased estimator since it takes all time steps into account when estimating the gradient. Even though the V2 rearranges the computational graph by the HO, it is still a biased estimator. This phenomenon indicates the importance of unbiasedness when conducting the fine-tuning task.

## 6 THEORETICAL PROPERTIES OF GRADIENT ESTIMATORS: BIAS, VARIANCE, AND CONVERGENCE

In this part, we analyze the bias of truncated BP and compare the variance of different estimators, backing the claim in Table 1. Thanks to the unbiasedness of the RLR estimator, the convergence of the optimization is also guaranteed.

To alleviate the memory burden of full-step BP, the truncated variant is often employed, backpropagating the gradient with only $T'$ steps; $T' \ll T$. However, the truncation introduces a **structural bias** into the gradient estimator. We have the following proposition to justify this structural bias.

**Proposition 6.1** (Biasedness of Truncated-BP ). *Assume $R$ and $\phi$ are differentiable almost everywhere, $R$ and $\phi_t$ have bounded gradients, then the FO estimator is unbiased. However, the truncated BP estimator $\nabla_\theta R(x_0)_{\text{truncated}}$ has a structural bias, which can be specified as below:*

$$\nabla_\theta \mathbb{E}[R(x_0)] - \mathbb{E}[\nabla_\theta R(x_0)_{\text{truncated}}] = \mathbb{E}_{z_{1:T}}\left[\left(\sum_{i=T'+1}^{T} \frac{\partial \phi_i(x_i, z_i; \theta)}{\partial \theta} \prod_{j=1}^{i-1} \frac{\partial x_{j-1}}{\partial x_j}\right)^\top \frac{dR(x_0)}{dx_0}\right].$$

Bias in the estimator can lead to suboptimal updates or even training failure, as the truncated gradient may not follow a true descent direction. This can cause two major issues: **model collapse** and **loss of multi-scale information**. In contrast, ZO (Spall, 1992) and HO (Jiang et al., 2024) are unbiased.

The stochastic nature of the DM results in the variance of the estimator. As expected, the variance of the FO estimator is lower than that of the HO and ZO estimators because the differentiation leverages the structural information of the neural network. However, BP introduces significant computational and storage overhead. The following proposition demonstrates that this additional cost is, to some extent, justified, as BP leverages accurate internal structures to reduce estimation variance.

**Proposition 6.2** (Variance Comparison). *Under Assumptions (A.1-3) in the Appendix, the variance of FO estimators is less than or equal to ZO estimators, i.e.*

$$\mathrm{Var}(\nabla_\theta R(x_0)) \leq \mathrm{Var}(R(x_0)\nabla \ln f(z)). \tag{9}$$

Based on the above proposition, it is straightforward to conclude that the variance of the HO estimator is also less than or equal to that of the ZO estimator, as it is essentially an FO estimator with a perturbation at the start of the sub-chain. The Table 1 presents all the gradient computation methods discussed above.

Overall, the RLR reorganizes the recursive computation chain by perturbation-based estimation, seamlessly integrating ZO, HO, and FO optimization techniques. RLR strikes a balance between computational cost and gradient accuracy, achieving both unbiasedness and low variance. The following Theorem 6.3 establishes its unbiasedness.

**Theorem 6.3** (Unbiasedness of RLR). *The RLR estimator is an unbiased estimator:*

$$
\begin{aligned}
\nabla_\theta \mathbb{E}[R(\phi_{1:T}(\boldsymbol{x}_T; \theta))] = \mathbb{E}_{z_{1:T}, j \sim \mathcal{U}(1, T-h)} \bigg[ & D_\theta^\top \phi_1(x_1, z_1; \theta) \frac{dR(x_0)}{dx_0} \\
& - R(x_0) D_\theta^\top \phi_{j:j+h}(x_{j+h}, z_{j:j+h}; \theta) \nabla_z \ln f(z_j) - \sum_{i \in C} R(x_0) \nabla_z \ln f(z_i). \bigg]
\end{aligned} \tag{10}
$$

The variance of the RLR estimator, denoted by $\sigma_{\mathrm{RLR}}^2$, is discussed in the appendix. Under limited computational resources where full BP is infeasible, RLR achieves substantially lower variance compared to other unbiased gradient estimators. Finally, the convergence rate of RLR is provided in the following Theorem 6.4.

**Theorem 6.4** (Convergence Rate). *Suppose that the reward function $R(\cdot)$ is L-smooth. By appropriately selecting the step size, the convergence rate of the RLR is given by*

$$\frac{1}{K+1} \sum_{k=0}^{K} \mathbb{E}(\|\nabla R(\theta_k)\|^2) \leq \sqrt{\frac{8L\Delta_0 \sigma_{\mathrm{RLR}}^2}{K+1}} + \frac{2L\Delta_0}{K+1},$$

*where $K$ is the number of iterations, $\theta_k$ is the trainable parameter in the $k$-th iteration, and $\Delta_0 = |R(\theta_0) - R^*|$ is difference between initialization performance and optimal performance.*

## 7 CONCLUSION

We propose the RLR optimizer, a half-order gradient estimation framework designed for efficient fine-tuning of diffusion models. Theoretically, we analyze the bias, variance, and convergence of the RLR estimator and formulate a constrained optimization problem to guide its design. Empirically, RLR consistently outperforms both reinforcement learning and truncated backpropagation methods on Text2Image and Text2Video tasks across multiple human preference reward models and benchmarks. Furthermore, we introduce a novel prompt technique, Diffusive Chain-of-Thought (DCoT), which complements the RLR and further boosts performance. Although determining the appropriate sub-chain length $h$ can be nontrivial in practice, we provide both theoretical justification and empirical ablations to guide practitioners in making informed choices.

## ACKNOWLEDGMENTS

This work was supported in part by the National Natural Science Foundation of China (NSFC) under Grants 72325007, and 72250065, and the Science and Technology Innovation Program of Hunan Province under Grant 2024RC7003.

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

APPENDIX CONTENTS

## A    EXTENDED RELATED WORKS

**Diffusion Probabilistic Models.**    Denoising Diffusion Model (Ho et al., 2020; Lu et al., 2022) is one of the strongest models for generation tasks, especially for visual generation (Rombach et al., 2022; Peebles & Xie, 2023; Chen et al., 2024a). Extensive research has been conducted from theoretical and empirical perspectives (Song et al., 2020a; Karras et al., 2022; Song et al., 2020b). It has achieved phenomenal success in multi-modality generation, including image, video, audio, and 3D shapes. The DM is trained on enormous images and videos from the internet (Bain et al., 2021; Wang et al., 2023b; Schuhmann et al., 2022). Empowered by modern architecture (Vaswani, 2017), it has a powerful learning capability for Pixel Space Distribution.

**Alignment and Post-training.**    After pre-training to learn the distribution of the targeted modality (Achiam et al., 2023; Kaplan et al., 2020), post-training is conducted to align the model toward specific preferences or tune the model to optimize a particular objective. RL has been utilized to align the foundation models toward various objectives (Ziegler et al., 2019; Lambert et al., 2022; Black et al., 2023). DPOK (Fan et al., 2024) studies KL regularization when training a separate DM for each prompt. Supervised learning can also be applied to the post-training phase (Rafailov et al., 2024), either optimizing an equivalent objective (Wallace et al., 2024) or directly differentiating the reward model (Clark et al., 2023; Prabhudesai et al., 2023; 2024). D3PO Yang et al. (2024) utilizes the DPO loss to train the DM. Specialist Diffusion Lu et al. (2023) focuses on sample-efficient, few-shot fine-tuning of large pre-trained diffusion models to enable the generation of new visual styles from as few as 5–10 examples. SPIN (Yuan et al., 2024b) introduces a self-play learning paradigm for diffusion models. For DM, most methods use a neural reward model to align the pre-trained model, and there has been a continual effort to design better reward models (He et al., 2024; Xu et al., 2024a;b).

**Forward Learning Methods.**    After extensive exploration of model training using forward inferences only (see, e.g., Peng et al., 2022; Hinton, 2022), forward-learning methods (Salimans et al., 2017; Malladi et al., 2023; Chen et al., 2023; Jiang et al., 2024) based on stochastic gradient estimation have recently emerged as a promising alternative to classical BP for large-scale machine learning problems (Zhao et al., 2024; Zhang et al., 2024a). Subsequent research (Ren et al., 2025; Chen et al., 2024b) has further optimized computational and storage overhead from various perspectives, achieving greater efficiency.

## B    EXPERIMENTS SETTINGS

### B.1    OVERALL SETTING

**Prompts dataset.**    We use Pick-a-Pic (Kirstain et al., 2023) and HPD v2 (Wu et al., 2023) for the Text2Image experiments. We report the performance of the trained model on unseen prompts from the training phase. For the Text2Video task, we prompt ChatGPT to generate a series of prompts that describe motions and train models under the prompts. After the training, we evaluate the model's performance on the unseen prompts from the VBench (Huang et al., 2024).

**Reward model and benchmark.**    We adopt multiple human preference reward models to train and test our methods, including PickScore (Kirstain et al., 2023), HPS v2 (Wu et al., 2023), and ImageReward (Xu et al., 2024b). All the models are trained on large-scale preference datasets. We also included the traditional aesthetic models, e.g., AES (Schuhmann, 2022).

For the video generation task, we use the VBench (Huang et al., 2024) to rate the methods according to various perspectives of the generated videos. We report 6 aspects of the generated videos in the main text: Subject Consistency (SC), Background Consistency (BC), Motion Smoothness (MS), Dynamic Degree (DD), Aesthetic Quality (AQ), and Imaging Quality (IQ). For more results of 16 metrics on VBench, please refer to Table 10 in the appendix.

**Baselines.**    We compared our methods with RL-based methods and BP-based methods. DDPO (Black et al., 2023) is the state-of-the-art RL method for DMs. For the Text2Image experiment, we include the AlignProp (Prabhudesai et al., 2023), a randomized truncated BP method. For

the Text2Video experiment, we included VADER (Prabhudesai et al., 2024), a BP-based method especially catering to video generation.

## B.2 PROMPTS

**HPD v2.** Human Preference Dataset v2 (HPD v2) is a large-scale collection designed to evaluate human preferences for images generated from text prompts, comprising 798,090 human-annotated preference choices from 433,760 image pairs. It includes images from diverse text-to-image models and utilizes cleaned prompts, processed with ChatGPT to remove biases and style-related words.

**Pick-a-Pic.** The Pick-a-Pic dataset is a publicly available collection of over half a million human preferences for images generated from 35,000 text prompts. Users generate images using state-of-the-art text-to-image models and select their preferred image from pairs, with each example including a prompt, two images, and a preference label. Collected via a web application, this dataset better reflects real-world user preferences and is used to train the PickScore scoring function, which enhances model evaluation and improvement.

**ChatGPT Created Prompts.** We ask ChatGPT to generate imaginative and detailed text descriptions for various scenarios, including people engaging in sports, animals wearing clothes, and animals playing musical instruments. We use the ChatGPT-generated prompts to train the model.

**Vbench Prompt Suite.** The Prompt Suite in VBench is a carefully curated set of text prompts designed to evaluate video generation models across 16 distinct evaluation dimensions. Each dimension is represented by approximately 100 prompts tailored to test specific aspects of video quality and consistency. The prompts are organized to reflect different categories, such as animals, architecture, human actions, and scenery, ensuring comprehensive coverage across diverse content types. These prompts are used to assess models' abilities, such as subject consistency, object class generation, motion smoothness, and more, providing insights into the models' strengths and weaknesses across various scenarios.

## B.3 REWARD MODELS AND EVALUATION METRICS

**PickScore.** The PickScore Reward Model is a scoring function trained on the Pick-a-Pic dataset, which includes human preferences for text-to-image generated images (Kirstain et al., 2023). It uses a CLIP-based architecture to compute scores by comparing text and image representations. Trained to predict user preferences, it minimizes KL-divergence between true preferences (preferred image or tie) and predicted scores.

**HPSv2.** Human Preference Score v2 (HPSv2) (Wu et al., 2023) is a model designed to evaluate human preferences for images generated by text-to-image models. Trained on the Human Preference Dataset v2, which includes 798,000 human preference annotations on 433,760 image pairs from various generative models, HPSv2 predicts which images are preferred based on text-image alignment and aesthetic quality, offering a more human-aligned evaluation compared to traditional metrics like Inception Score or Fréchet Inception Distance.

**AES.** The Aesthetic Score (AES) (Schuhmann, 2022) is obtained from a model that builds on CLIP embeddings and incorporates additional multilayer perceptron (MLP) layers to capture the visual attractiveness of images. This metric serves as a tool for assessing the aesthetic quality of generated images, offering insights into how closely they match human aesthetic preferences.

**ImageReward.** ImageReward (Xu et al., 2024b) is a reward model designed to evaluate human preferences in text-to-image generation. It is trained on a large dataset of 137k expert comparisons, using a systematic annotation pipeline that rates and ranks images based on alignment with text, fidelity, and harmlessness. Built on the BLIP model, ImageReward accurately predicts human preferences, outperforming models like CLIP, Aesthetic, and BLIP. It serves as a promising automatic evaluation metric for text-to-image synthesis, aligning well with human rankings.

## B.4 BASELINES

**DOODL.** DOODL (Direct Optimization of Diffusion Latents) optimizes diffusion latents to improve image generation by directly adjusting latents based on a model-based loss. Unlike traditional

classifier guidance methods, DOODL avoids the need for retraining models or using approximations, providing more accurate and efficient guidance. It enhances text-conditioned generation, expands pre-trained model vocabularies, enables personalized image generation, and improves aesthetic quality, offering better control and higher-quality outputs in generative image models.

**DDPO.** DDPO (Denoising Diffusion Policy Optimization) is an RL-based method for optimizing diffusion models towards specific goals like image quality or compressibility. By treating denoising as a multi-step decision-making task, DDPO uses policy gradients to maximize a reward function, unlike traditional likelihood-based methods. DDPO also shows strong generalization across diverse prompts, making it highly effective for fine-tuning generative models.

**AlignProp.** AlignProp fine-tunes text-to-image diffusion models by backpropagating gradients through the entire denoising process using randomized truncated backpropagation. This method reduces memory and computational costs by employing low-rank adapter modules and gradient checkpointing. The randomized TBTT approach, which randomly selects the number of backpropagation steps, prevents overfitting and mode collapse, improving both sample efficiency and reward optimization. AlignProp outperforms other methods in terms of generalization, image-text alignment, and aesthetic quality, making it a highly efficient and effective tool for optimizing diffusion models to specific downstream objectives.

**VADER.** VADER (Video Diffusion via Reward Gradients) fine-tunes video diffusion models by backpropagating gradients from pre-trained reward models. It enhances sample and computational efficiency, using reward models to assess aesthetics, text-video alignment, and other video-specific tasks. VADER maintains temporal consistency and generalizes well to unseen prompts, making it an effective tool for adapting video models to complex objectives.

### B.5 ORTHOGONAL TRICKS

**LoRA** applies low-rank adaptation to the original parameters, $\theta$, by fine-tuning only the low-rank components rather than the full parameters. Specifically, each linear layer in the backbone (U-Net or Transformer) of the diffusion model is modified as $h = Wx + BAx$, where $W \in \mathbb{R}^{m \times m}$, $A \in \mathbb{R}^{m \times k}$, and $B \in \mathbb{R}^{k \times m}$, with $k \ll m$. The LoRA weights are initialized to zero, ensuring no initial impact on the pre-trained model's performance. This method reduces the number of parameters to be trained while achieving performance comparable to full-parameter fine-tuning. We apply LoRA with $k = 16$ to all experiments.

**Gradient checkpointing** is a well-known technique for reducing memory usage during neural network training (Gruslys et al., 2016; Chen et al., 2016). Instead of storing all intermediate activations for backpropagation, it selectively saves only those needed for gradient computation and transfers the rest to the CPU's main memory. However, this comes with the cost of additional data transfer and computation overhead, which can increase training time. In the case of truncated backpropagation, checkpointing is unavoidable. For our RLR optimizer, though, gradient checkpointing is not a necessary technique.

## C HYPEPARAMETERS

All the experiments are conducted on a machine with 8 NVIDIA V100 GPUs. Each GPU has 32GB of memory.

For the Text2Image and the DCoT experiment, we use Adam optimizer with the learning rate of $5 \times 10^{-4}$. The batch size is $4$ and the gradient accumulation steps are $2$. The DDIM steps are $50$ and the classifier guidance weight is 7.5. The local sub-chain has a length of $2$. We use Gaussian noise with a standard deviation of $1 \times 10^{-3}$ for perturbing the parameters.

For the Text2Video experiment, we use Adam optimizer with the learning rate of $1 \times 10^{-4}$. The batch size is $1$ and the gradient accumulation steps are $8$. The DDIM steps are $25$ and the classifier guidance weight is 7.5. The local sub-chain has a length of $2$. We use Gaussian noise with a standard deviation of $1 \times 10^{-4}$ for perturbing the parameters.

# D    Qualitative Results of Text2Image

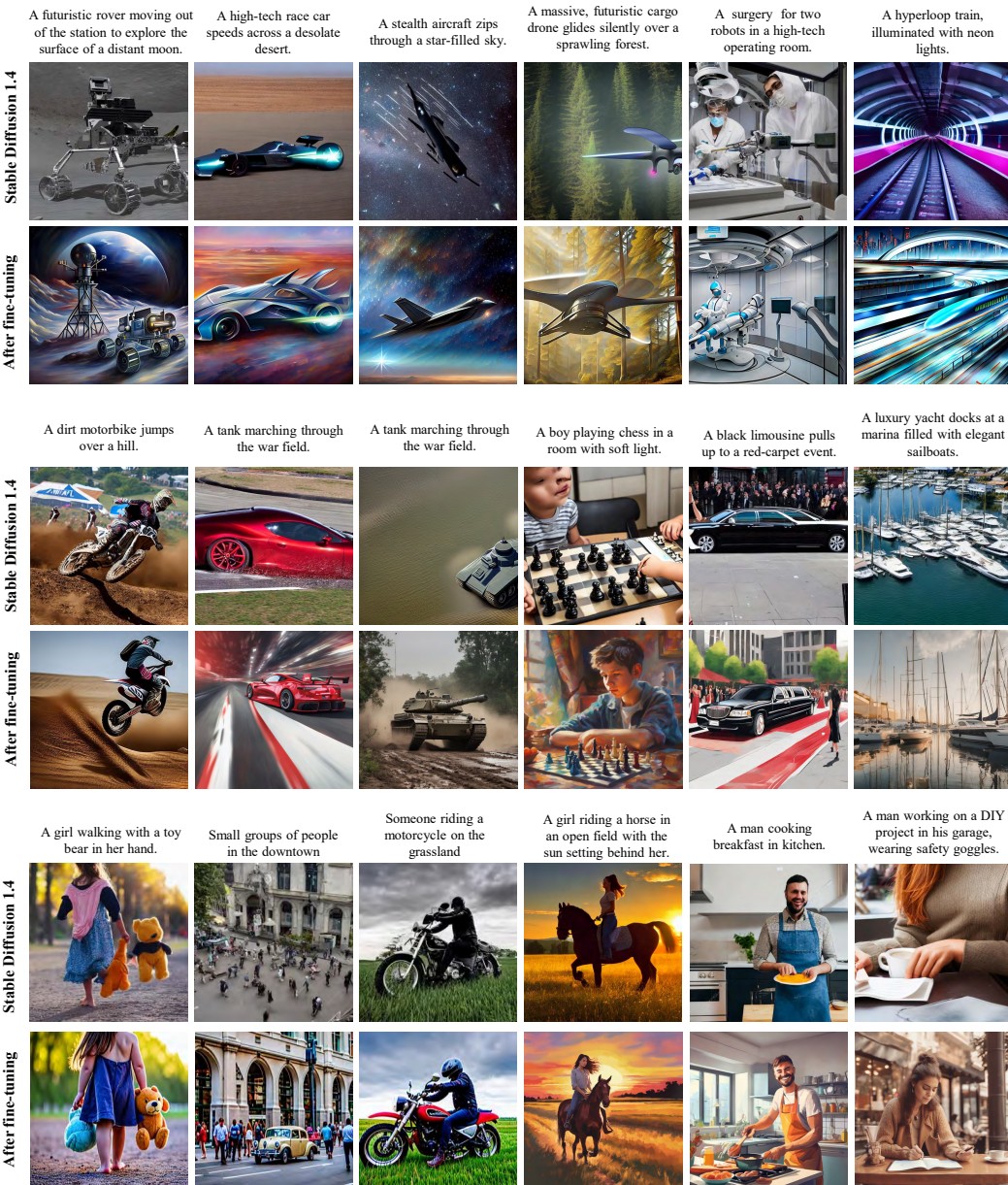

Figure 7: Qualitative results for Text2Image generation.

# E    QUALITATIVE RESULTS OF TEXT2VIDEO

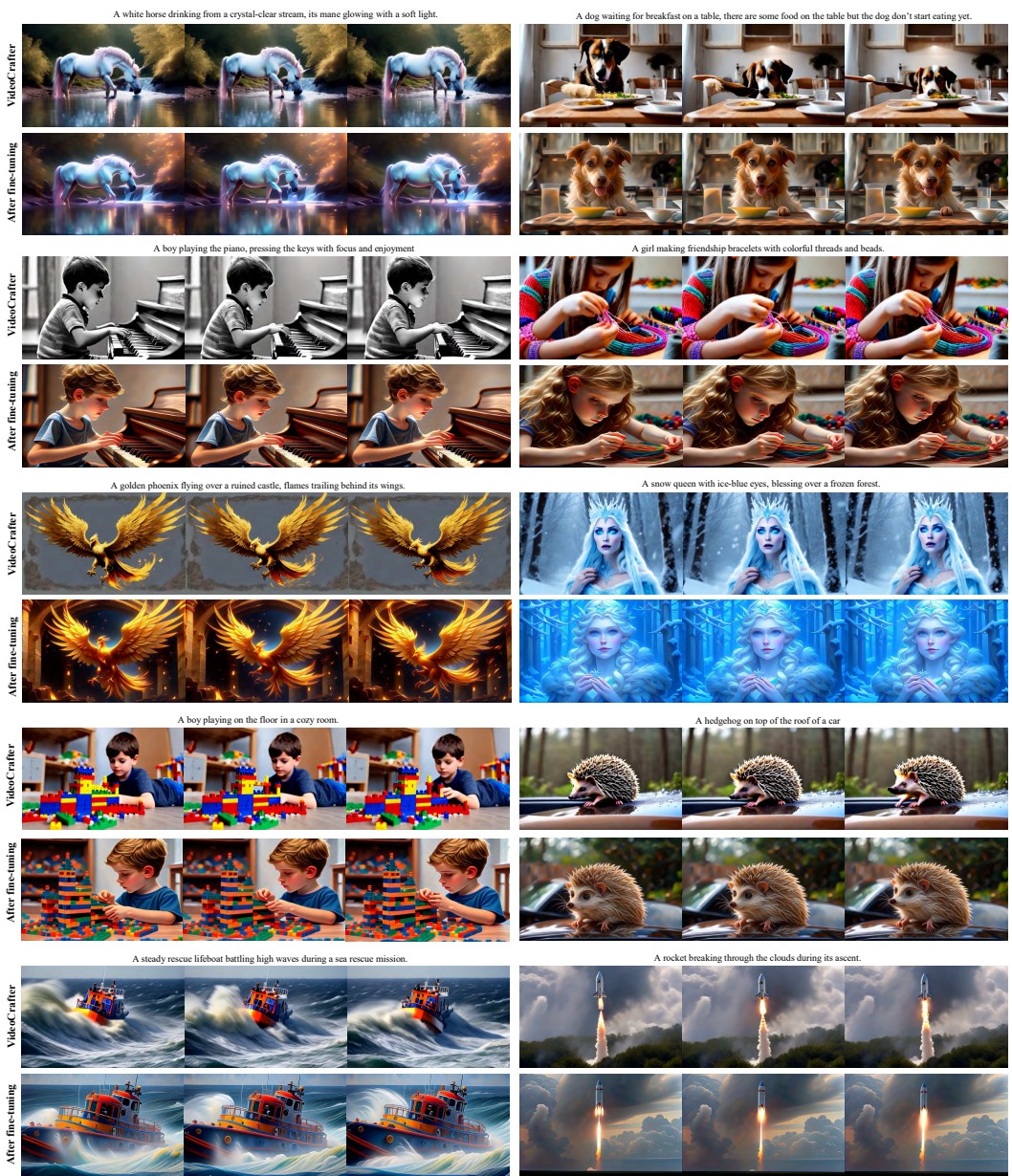

Figure 8: Qualitative results for Text2Video generation.

## F  DETAILS FOR DIFFUSIVE CHAIN-OF-THOUGHT EXPERIMENT

Original prompts for the hand task: (1) A realistic open palm facing upward. (2) A picture of a hand facing downward. (3) A hand in a relaxed position. (4) A hand. (5) A photo of a hand. The generated DCoT prompts are in Figure 10. We show the instructions to generate the multi-scale DCoT prompts in Figure 11.

Stable Diffusion 1.4                    After fine-tuning by RLR with DCoT

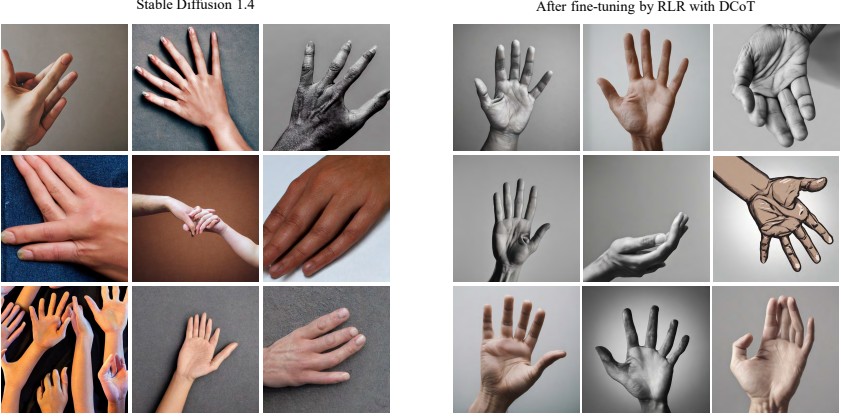

Figure 9: Qualitative results for the hand task in DCoT.

| A realistic open palm facing upward | |
|---|---|
| **Coarse-level** | A single human hand positioned with the palm facing upward |
| **Mid-level** | An open palm with fingers naturally spread and relaxed, oriented upward |
| **Fine-level** | Detailed skin texture, visible creases on the palm, and natural lighting on fingernails and knuckles |

| A picture of a hand facing downward | |
|---|---|
| **Coarse-level** | A human hand with the palm facing downward |
| **Mid-level** | Fingers extended naturally, back of the hand visible, wrist slightly relaxed |
| **Fine-level** | Visible veins, knuckle contours, skin texture, and soft shadows on the fingers |

| A hand in a relaxed position | |
|---|---|
| **Coarse-level** | A human hand resting in a neutral, relaxed position |
| **Mid-level** | Fingers slightly curved, palm partially visible, wrist aligned naturally |
| **Fine-level** | Subtle skin folds at the joints, soft lighting on the knuckles, natural skin tone and texture |

Figure 10: Qualitative examples for DCoT prompts

---
**Instruction to generate the DCoT prompts**

Given an original prompt for visual generation, you need to break it down into three prompts that controls different scales of the visual content: coarse-scale, mid-scale, fine-scale. The generated prompts should be concise and clear.

Here is the original prompt:
<PROMPT>

Your answer format should be:
Coarse-level:
<COMPLETE>

Mid-level:
<COMPLETE>

Fine-level:
<COMPLETE>

---

Figure 11: Instruction to generate DCoT prompts.

## G  MEMORY PROFILE, TIME COMPLEXITY, AND DIFFERENT DIFFUSION SOLVERS

We give the memory cost for the experiments with Text2Image on SD 1.4 in Table 6. We offload the memory to the CPU RAM to avoid the out-of-memory error. The AlignProp based on truncated BP has the largest memory consumption. The DDPO, based on RL, has the smallest consumption, while the sample efficiency is terrible, as shown in Figure 5. Our RLR has significantly lower memory consumption than the AlignProp.

Table 6: Memory cost of Text2Image experiments on SD 1.4

| Methods | VRAM | System RAM | Total |
|---|---|---|---|
| DDPO | 12.4 GB | 0 GB | 12.4 GB |
| AlignProp | 25.4 GB | 78.5 GB | 103.9 GB |
| RLR | 22.4 GB | 0 GB | 22.4 GB |

We report the per-step time cost for the truncated BP method, AlignProp, and compare the three methods in Table 7. As shown in the following table, AlignProp has the highest per-step cost, while RL is the fastest. When considering the total time to reach the same performance level (AES = 5.4), the advantage of RLR becomes more evident. RL requires significantly more steps to converge due to high-variance gradients, whereas RLR achieves the target in fewer steps thanks to its low-variance property, resulting in faster convergence than both RL and AlignProp.

Table 7: Time complexity of Text2Image experiments on SD 1.4

| Methods | RLR | RL | Alignprop |
|---|---|---|---|
| Time(min/step) | 1.61 | 0.82 | 2.85 |
| Time(to same score) | 121 (75 steps) | 492 (600 steps) | 285 (100 steps) |

We use RLR to train SD1.4 with the DPMSolverMultistepScheduler to verify its effectiveness on different samplers. We set the inference steps to 20 and the solver order to 2. The results in the Table 8 show that RLR maintains robust performance across both samplers. Diffusion-based generation methods such as DDIM are deterministic when $\eta = 0$, and only become stochastic when $\eta > 0$. In our experiments, when we require the HO estimator at a given step, we explicitly set $\eta = 0.1$ for the step to perform HO. For all other steps, we retain the deterministic ODE solver structure by setting

$\eta = 0$. This approach allows us to flexibly combine deterministic and stochastic transitions within the same framework, ensuring the correctness of the HO estimator where needed. DPM-Solver supports both deterministic and stochastic sampling modes. Stochasticity can be selectively enabled at the specific steps where the HO estimator is used, with all other steps remaining deterministic.

Table 8: Ablation of different diffusion solver in Text2Image experiments on SD 1.4

| Methods | PickScore | HPSv2 | AES | ImageReward |
|---------|-----------|-------|-----|-------------|
| DDIM | 20.14 | 28.57 | 6.53 | 75.65 |
| DPMSolver | 20.21 | 28.55 | 6.56 | 75.70 |

## H   ABLATIONS ON THE SUB-CHAIN LENGTH $h$

Table 9: Comparison of methods on HPSv2 and ImageReward with memory(GB) and time cost(minute per step).

| Method | HPSv2 | ImageReward | Memory | Wall clock time |
|--------|-------|-------------|--------|-----------------|
| ZO | 22.31 | 40.82 | 11.8 | 0.79 |
| DDPO (RL) | 22.79 | 52.06 | 12.4 | 0.82 |
| RLR($h = 0$) | 26.70 | 63.85 | 12.7 | 0.90 |
| RLR($h = 1$) | 28.02 | 69.85 | 18.8 | 1.15 |
| RLR($h = 2$) | 29.22 | 76.55 | 22.4 | 1.61 |
| RLR($h = 3$) | 29.36 | 76.62 | 32.6 | 5.65 |
| RLR($h = 4$) | 29.55 | 76.70 | 45.8 | 9.23 |

We provide ablation over different sub-chain lengths $h$ and different estimators in the following two tables. We report the reward score from two models, memory, and wall clock time. As the $h$ increases, the performance increases due to the reduced variance. Increasing $h$ would also increase the computation overhead, including memory and time. We find that when $h$ is larger than 2, the performance gain is marginal compared to the increasing computation burden (high memory requirement and long wall clock time). When the $h$ is smaller than 2, high variance would lead to performance degradation. Therefore, we choose $h = 2$ as the best hyperparameter in the main experiment.

## I   MORE RESULTS ON VBENCH

**VBench Evaluation Dimensions.**   VBench evaluates video generation models across 16 disentangled dimensions, categorized into **Quality** and **Semantic** groups.

The **Quality** category consists of 7 metrics: **Subject Consistency** and **Background Consistency** measure identity and scene consistency across frames using feature similarity; **Motion Smoothness** evaluates physically plausible motion via a motion prior model; **Dynamic Degree** quantifies motion magnitude to penalize overly static videos; **Temporal Flickering** measures high-frequency temporal instability; **Aesthetic Quality** reflects perceptual appeal using an aesthetic predictor; and **Imaging Quality** assesses distortions like blur or noise using an image quality model.

The **Semantic** category includes 9 metrics: **Object Class** and **Multiple Objects** test object presence and compositionality; **Human Action** verifies accurate motion execution; **Color**, **Spatial Relationship**, and **Scene** check fidelity to prompt-specified attributes and layouts; **Appearance Style** and **Temporal Style** assess stylistic alignment in space and time; and **Overall Consistency** captures general text-video correspondence. Each dimension has tailored prompts and automatic evaluation pipelines, ensuring fine-grained, human-aligned assessment.

According to the VBench protocol, the **Total Score**(TS) is computed as a weighted sum of the **Quality Score**(QS) and **Semantic Score**(SS), following the formula: TS $= 0.8 \cdot$ QS $+ 0.2 \cdot$ SS. In our evaluation, the proposed RLR method achieves the highest scores across all three levels: QS, SS,

and the final TS. Notably, most metrics contributing heavily to QS are reported in the main text due to their strong correlation with perceptual quality and their larger weights in the TS calculation. For completeness, we present all 16 VBench metrics in Table 10, where RLR consistently outperforms existing baselines.

Table 10: Automatic evaluation on VBench.

(a) Quality dimensions and total score.

| Method | Subject Consistency | Background Consistency | Motion Smoothness | Dynamic Degree | Aesthetic Quality | Imaging Quality | Temporal Flickering | Quality Score | Total Score |
|---|---|---|---|---|---|---|---|---|---|
| ModelScopeT2V | 89.97 | 89.87 | 95.79 | 66.39 | 52.06 | 58.57 | 98.28 | 78.05 | 75.75 |
| Open-Sora | 92.09 | 97.39 | 95.61 | 48.61 | 57.76 | 61.51 | 98.41 | 78.82 | 75.91 |
| Pika | 96.76 | 98.95 | 99.51 | 37.22 | 63.15 | 62.33 | 99.77 | 82.68 | 80.40 |
| Gen-2 | 97.61 | 97.61 | 99.58 | 18.89 | 66.96 | 67.42 | 99.56 | 82.47 | 80.58 |
| T2V-Turbo | 96.28 | 97.02 | 97.34 | 49.17 | 63.04 | 72.49 | 97.48 | 82.57 | 81.01 |
| DDPO | 95.53 | 96.63 | 96.92 | 58.29 | 59.23 | 66.84 | 97.63 | 81.43 | 79.84 |
| VADER | 95.79 | 96.71 | 97.06 | 66.94 | 66.04 | 69.93 | 97.62 | 83.75 | 81.84 |
| RLR | 97.64 | 97.19 | 98.05 | 70.69 | 66.15 | 71.08 | 97.70 | **85.20** | **83.14** |

(b) Semantic dimensions.

| Method | Object Class | Multiple Objects | Human Action | Color | Spatial Relationship | Scene | Appearance Style | Temporal Style | Overall Consistency | Semantic Score |
|---|---|---|---|---|---|---|---|---|---|---|
| ModelScopeT2V | 82.25 | 38.98 | 92.40 | 81.72 | 33.68 | 39.26 | 23.39 | 25.37 | 25.67 | 66.54 |
| Open-Sora | 74.98 | 33.64 | 85.00 | 78.15 | 43.95 | 37.33 | 21.58 | 25.46 | 26.18 | 64.28 |
| Pika | 87.45 | 46.69 | 88.00 | 85.31 | 65.65 | 44.80 | 21.89 | 24.44 | 25.47 | 71.26 |
| Gen-2 | 90.92 | 55.47 | 89.20 | 89.49 | 66.91 | 48.91 | 19.34 | 24.12 | 26.17 | 73.03 |
| T2V-Turbo | 93.96 | 54.65 | 95.20 | 89.90 | 38.67 | 55.58 | 24.42 | 25.51 | 28.16 | 74.76 |
| DDPO | 91.36 | 45.67 | 94.81 | 90.22 | 37.54 | 55.37 | 25.27 | 25.04 | 28.03 | 73.47 |
| VADER | 91.89 | 48.67 | 95.12 | 91.56 | 39.78 | 54.39 | 25.14 | 25.23 | 28.19 | 74.21 |
| RLR | 92.52 | 50.22 | 95.20 | 92.31 | 40.33 | 54.18 | 25.74 | 25.35 | 28.27 | **74.87** |

## J  PROOFS

### J.1  PROOF OF PROPOSITION 6.1

*Proof.* We assume: $R : \mathbb{R}^d \to \mathbb{R}$ and $\phi : \mathbb{R}^d \times \mathbb{R}^m \times \Theta \to \mathbb{R}^d$ are continuously differentiable, and their gradients with respect to the targeted arguments are uniformly bounded. That is, there exist finite constants $M_R > 0$ and $M_\phi > 0$ such that $\sup_x |\nabla R(x)| \le M_R$, $\sup_{(x_t, z_t, \theta)} |\nabla_{(x_t, \theta)} \phi_t(x_t, z_t, \theta)| \le M_\phi$. Under these assumptions, for the composite function $R(\phi_{1:T}(x_T, z_{1:T}, \theta))$, the partial derivative with respect to $\theta$ is also uniformly bounded by the chain rule (as a product of bounded terms). Therefore, the integrability condition required by the Dominated Convergence Theorem (Rudin, 1987) is satisfied, which justifies interchanging the gradient and the expectation (Glasserman, 1990).

By chain rule and $x_0 = \phi_{1:T}(x_T, z_{1:T}; \theta) = \phi_1(x_1, z_1; \theta)$, we have

$$
\frac{\partial R(x_0)}{\partial \theta} = \frac{\partial x_0}{\partial \theta}^\top \frac{dR(x_0)}{dx_0} = \left[ \frac{\partial \phi_1(x_1, z_1; \theta)}{\partial \theta} + \frac{\partial x_0}{\partial x_1} \frac{\partial x_1}{\partial \theta} \right]^\top \frac{dR(x_0)}{dx_0}
$$

$$
= \left[ \frac{\partial \phi_1(x_1, z_1; \theta)}{\partial \theta} + \frac{\partial x_0}{\partial x_1} \left[ \frac{\partial \phi_2(x_2, z_2; \theta)}{\partial \theta} + \frac{\partial x_1}{\partial x_2} \frac{\partial x_2}{\partial \theta} \right] \right]^\top \frac{dR(x_0)}{dx_0}
$$

$$
= \ldots
$$

$$
= \left[ \frac{\partial \phi_1(x_1, z_1; \theta)}{\partial \theta} + \sum_{i=2}^{T} \frac{\partial \phi_i(x_i, z_i; \theta)}{\partial \theta} \prod_{j=1}^{i-1} \frac{\partial x_{j-1}}{\partial x_j} \right]^\top \frac{dR(x_0)}{dx_0}.
$$

Therefore, we can reach the conclusion that

$$
\nabla_\theta \mathbb{E}_z [R(x_0)] = \mathbb{E}_{z_{1:T}} [\nabla_\theta R(\phi_1(x_1, z_1; \theta))]
$$

$$
= \mathbb{E}_{z_{1:T}} \left[ \left[ \frac{\partial \phi(x_1, z_1; \theta)}{\partial \theta} + \sum_{i=2}^{T} \frac{\partial \phi_i(x_i, z_i; \theta)}{\partial \theta} \prod_{j=1}^{i-1} \frac{\partial x_{j-1}}{\partial x_j} \right]^\top \frac{dR(x_0)}{dx_0} \right],
$$

which means the unbiasedness of the FO estimator. Furthermore, the truncated BP estimator is

$$\nabla_\theta R(\boldsymbol{x}_0)_{\text{truncated}} = \left[ \frac{\partial\phi_1(x_1, z_1; \theta)}{\partial\theta} + \sum_{i=2}^{T'} \frac{\partial\phi_i(x_i, z_i; \theta)}{\partial\theta} \prod_{j=1}^{i-1} \frac{\partial x_{j-1}}{\partial x_j} \right]^\top \frac{dR(x_0)}{dx_0}. \quad (11)$$

The structural bias of the truncated BP estimator can be specified by combining the above results:

$$\nabla_\theta \mathbb{E}[R(x_0)] - \mathbb{E}[\nabla_\theta R(x_0)_{\text{truncated}}] = \mathbb{E}_{z_{1:T}}\left[ \sum_{i=T'+1}^{T} \frac{\partial\phi(x_i; \theta)}{\partial\theta} \prod_{j=1}^{i-1} \frac{\partial\phi(x_j; \theta)}{\partial x_j} \right]^\top \frac{dR(x_0)}{dx_0},$$

which completes the proof. □

### J.2   PROOF OF PROPOSITION 6.2 AND ADDITIONAL ASSUMPTIONS

**Assumption J.1.** Define $R(z; \theta)$ as $R(\phi_{1:T}(x_T, z_{1:T}; \theta))$. Assume that $R(z; \theta)$ is differentiable with respect to $\theta$ almost surely, $\mathbb{P}(R(\theta) = r) = 0$ for every $r \in \mathbb{R}$, and Lipschitz condition holds for every $\theta_1$ and $\theta_2$:
$$|R(z; \theta_1) - R(z; \theta_2)| \leq m_1(z)|\theta_1 - \theta_2|,$$
where $m_1(z)$ is integrable.

**Assumption J.2.** For any $x_t$, whose randomness comes from $z$, the density $f(x_t; \theta)$ is differentiable with respect to $\theta$, and uniform integrability holds:
$$\sup_\theta \left| R(z; \theta) \frac{\partial}{\partial\theta} f(x_t; \theta) \right| \leq m_2(z),$$
where $m_2(z)$ is integrable.

**Assumption J.3.** $R(z; \theta)$ is twice continuously differentiable. The following functions are integrable: $m_1(\cdot)^2, m_2(\cdot)^2, \sup_\theta |R(\cdot; \theta)| \times m_1(\cdot), \sup_\theta |R(\cdot; \theta)| \times \sup_\theta |R''(\cdot; \theta)|$.

*Proof of Proposition 6.2.* Since $R(z; \theta)$ is the reward function and the random variables $z_{1:T}$ are Gaussian distributions in our case, it is easy to check that the above assumptions are satisfied. By applying Theorem 2 in Cui et al. (2020), we can reach the conclusion. □

### J.3   PROOF OF THEOREM 6.3

*Proof.* The RLR estimator contains three parts: FO estimator terms, HO estimator terms, and ZO estimator terms. For simplicity, we can consider the terms with an FO estimator term, an HO estimator term, and a ZO estimator term. Substituting the specific form of the iteration process, we have

$$x_0 = \phi(x_1, z_1; \theta), \quad x_1 = \varphi(x_2; \theta + z_2), \quad x_2 = \varphi(x_3; \theta) + z_3,$$

where $x_1 = \varphi(x_2; \theta + z_2)$ is an ZO estimator term and $x_2 = \varphi(x_3; \theta) + z_3$ is an HO estimator term. We define $z_2 \sim f_{\text{ZO}}(z)$ and $z_3 \sim f_{\text{HO}}(z)$ to indicate the noise distribution associated with ZO and HO estimators. The overall gradient can be written as:

$$\nabla_\theta \mathbb{E}_{z_{1:3}}[R(x_0)] = \sum_{i=1}^{3} \nabla_{\theta_i} \mathbb{E}_{z_{1:3}}[R(\phi(\varphi(\varphi(x_3; \theta_3) + z_3; \theta_2 + z_2), z_1; \theta_1))] \Big|_{\theta_1=\theta_2=\theta_3=\theta} \quad (12)$$

For the FO term, $\phi(x_1, z_1; \theta)$, the gradient is derived by applying the chain rule:

$$\nabla_{\theta_1} \mathbb{E}_{z_{1:3}}[R(\phi(\varphi(\varphi(x_3; \theta) + z_3; \theta + z_2), z_1; \theta_1))] \Big|_{\theta_1=\theta}$$

$$= \nabla_{\theta_1} \mathbb{E}_{z_{1:3}}[R(\phi(x_1, z_1; \theta_1))] \Big|_{\theta_1=\theta} = \mathbb{E}_{z_{1:3}}\left[ D_{\bar{\theta}}^\top \phi(x_1, z_1; \bar{\theta}) \frac{dR(x_0)}{dx_0} \right] \Big|_{\bar{\theta}=\theta}$$

To derive the gradient of the ZO and HO terms, the key idea is to push the parameter of interest into the density function of the corresponding noise via a change of variables, so that the gradient operator only acts on the log-density function. We have the gradient of the ZO term:

$$\nabla_{\theta_2}\mathbb{E}_{z_1,z_2,z_3}[R(\phi(\varphi(\varphi(x_3;\theta)+z_3;\theta_2+z_2),z_1;\theta)))]\Big|_{\theta_2=\theta}$$

$$= \nabla_{\theta_2}\mathbb{E}_{z_1,z_2,z_3}[R(\phi(\varphi(x_2;\theta_2+z_2),z_1;\theta)))]\Big|_{\theta_2=\theta}$$

$$= \nabla_{\theta_2}\mathbb{E}_{z_1,z_3}[\mathbb{E}_{z_2\sim f_{ZO}(z)}[R(\phi(\varphi(x_2;\theta_2+z_2),z_1;\theta))|z_1,z_3]]\Big|_{\theta_2=\theta}$$

$$= \nabla_{\theta_2}\mathbb{E}_{z_1,z_3}[\mathbb{E}_{v_2\sim f_{ZO}(v-\theta_2)}[R(\phi(\varphi(x_2;v_2),z_1;\theta))|z_1,z_3]]\Big|_{\theta_2=\theta}$$

$$= \mathbb{E}_{z_1,z_3}[\mathbb{E}_{v_2\sim f_{ZO}(v-\theta_2)}[R(x_0)\nabla_{\theta_2}\ln f_{ZO}(v_2-\theta_2)|z_1,z_3]]\Big|_{\theta_2=\theta}$$

$$= \mathbb{E}_{z_1,z_3}[\mathbb{E}_{z_2\sim f_{ZO}(z)}[-R(x_0)\nabla_z\ln f_{ZO}(z_2)|z_1,z_3]]\Big|_{\theta_2=\theta}$$

$$= \mathbb{E}_{z_1,z_2,z_3}[-R(x_0)\nabla_z\ln f_{ZO}(z_2)],$$

where the first equality follows from the tower property of expectations, the second from a change of variables, i.e., $v_2 = \theta + z_2$, the third uses the likelihood-ratio trick, and the fourth substitutes the variable back and collapses the expectation. Likewise, with the change of variables $v_3 = \varphi(x_3;\theta_3) + z_3$, the derivation for the HO term follows analogously, yielding:

$$\nabla_{\theta_3}\mathbb{E}_{z_1,z_2,z_3}[R(\phi(\varphi(\varphi(x_3;\theta_3)+z_3;\theta+z_2),z_1;\theta)))]\Big|_{\theta_3=\theta}$$

$$= \nabla_{\theta_3}\mathbb{E}_{z_1,z_2}[\mathbb{E}_{z_3\sim f_{HO}(z)}[R(\phi(\varphi(\varphi(x_3;\theta_3)+z_3;\theta+z_2),z_1;\theta))|z_1,z_2]]\Big|_{\theta_3=\theta}$$

$$= \nabla_{\theta_3}\mathbb{E}_{z_1,z_2}[\mathbb{E}_{v_3\sim f_{HO}(v-\varphi(x_3;\theta_3))}[R(\phi(\varphi(v_3;\theta+z_2),z_1;\theta))|z_1,z_2]]\Big|_{\theta_3=\theta}$$

$$= \mathbb{E}_{z_1,z_2}[\mathbb{E}_{v_3\sim f_{HO}(v-\varphi(x_3;\theta_3))}[R(x_0)\nabla_{\theta_3}\ln f_{HO}(v_3-\varphi(x_3;\theta_3))|z_1,z_2]]\Big|_{\theta_3=\theta}$$

$$= \mathbb{E}_{z_1,z_2}\left[\mathbb{E}_{z_3\sim f_{HO}(z)}[-R(x_0)D_{\theta_3}^\top\varphi(x_3;\theta_3)\nabla_z\ln f_{HO}(z_3)|z_1,z_2]\right]\Big|_{\theta_3=\theta}$$

$$= \mathbb{E}_{z_1,z_2,z_3}\left[-R(x_0)D_{\bar\theta}^\top\varphi(x_3;\bar\theta)\nabla_z\ln f_{HO}(z_3)\right]\Big|_{\bar\theta=\theta}.$$

Finally, we have the unbiased RLR gradient estimator:

$$\nabla_\theta\mathbb{E}_{z_{1:3}}[R(x_0)] = \mathbb{E}_{z_{1:3}}\left[\frac{\partial\phi(x_1,z_1;\bar\theta)}{\partial\bar\theta}^\top\frac{dR(x_0)}{dx_0} - R(x_0)\left(\nabla_z\ln f_{ZO}(z_2) + \frac{\partial\varphi(x_3;\bar\theta)}{\partial\bar\theta}^\top\nabla_z\ln f_{HO}(z_3)\right)\right]\Big|_{\bar\theta=\theta}.$$

Since the sum of unbiased estimators is still unbiased, it is easy to generalize the result for any number of terms along the lines of the above proof. Also, the unbiasedness of the estimators does not change no matter how we pick the combination of the HO estimator and the ZO estimator. Therefore, we can reach the conclusion that:

$$\nabla_\theta\mathbb{E}_{\boldsymbol{z}_{1:3}}[R(\boldsymbol{x}_0)] = D_\theta^\top\phi(x_1,z_1;\theta)\frac{dR(x_0)}{dx_0}$$

$$- R(x_0)D_\theta^\top\phi_{j:j+h}(x_{j+h},z_{j:j+h};\theta)\nabla_z\ln f(z_j) - \sum_{i\in C}R(x_0)\nabla_z\ln f(z_i),$$

which completes the proof. □

### J.4 VARIANCE OF THE RLR ESTIMATOR

In this section, we discuss the variance of the RLR estimator, which consists of 3 terms, which are denoted as $A$, $B$, and $C$, respectively:

$$RLR = \underbrace{\frac{\partial \phi_1(x_1, z_1; \theta)}{\partial \theta}}_{A} - \underbrace{R(x_0) D_\theta^\top \phi_{j:j+h}(x_{j+h}, z_{j:j+h}; \theta) \nabla \ln f(z_j)}_{B} - \underbrace{\sum_{i \in C} R(x_0) \nabla \ln f(z_i)}_{C}.$$

(13)

It is easy to verify that the variance of RLR can be bounded by the variances of terms A, B and C:

$$\text{Var}(RLR) = \text{Var}(A + B + C)$$
$$= \text{Var}(A) + \text{Var}(B) + \text{Var}(C) + 2 \text{Cov}(A, B) + 2 \text{Cov}(A, C) + 2 \text{Cov}(B, C)$$
$$\leq \text{Var}(A) + \text{Var}(B) + \text{Var}(C) + 2 \left( \sqrt{\text{Var}(A)\text{Var}(B)} + \sqrt{\text{Var}(A)\text{Var}(C)} + \sqrt{\text{Var}(B)\text{Var}(C)} \right),$$

(14)

which gives an upper bound for the variance of the RLR estimator. Next we derive the specific expression of the terms $\text{Var}(A)$, $\text{Var}(B)$ and $\text{Var}(C)$.

First, the term $A$ represents the original exact BP without any injected noise, so its variance is 0. Next, for the terms $B$ and $C$, since both are LR estimators, we present them in the following unified form for simplicity:

$$\eta = \mathbb{E}[R(z) \nabla_\theta \ln f(z, \theta)],$$

where $\nabla_\theta \ln f(z, \theta) = D_\theta^\top \phi_{j:j+h}(\boldsymbol{x}_{j+h}; \theta) \nabla \ln f(\boldsymbol{z}_j)$ in the term B. The variance of $\eta$ is given by

$$\text{Var}(\eta) = \mathbb{E}[(R(z) \nabla_\theta \ln f(z, \theta))^2] - \mu(\theta)^2,$$

where $\mu(\theta)$ is the estimator mean.

To better characterize the variance, we now derive an alternative form of the gradient:

$$\mathbb{E}[R(z) \nabla_\theta \ln f(z, \theta)] = \int \lim_{h \to 0} \frac{f(z; \theta + h) - f(z; \theta)}{h} R(z) dz$$
$$= \lim_{h \to 0} \int \frac{f(z; \theta + h) - f(z; \theta)}{h} R(z) dz$$
$$= \lim_{h \to 0} \frac{1}{h} \int f(z; \theta) \left( \frac{f(z; \theta + h)}{f(z; \theta)} - 1 \right) R(z) dz$$
$$= \lim_{h \to 0} \frac{1}{h} (\mathbb{E}[\omega(\theta, h) R(z)] - \mathbb{E}_f[R(z)]),$$

(15)

where the importance weight $\omega(\theta, h) = \frac{f(z; \theta + h)}{f(z; \theta)}$.

With this alternative form Equation (15), we have the variance of the LR estimator

$$\text{Var}(\eta) = \lim_{h \to 0} \mathbb{E}[(w(\theta, h) - 1)^2 R(z)^2] - \mu(\theta)^2,$$

(16)

By the Hammersley-Chapman-Robbins bound, we can derive a lower bound for the variance:

$$\text{Var}(\eta) \geq \sup_h \frac{(\mu(\theta + h) - \mu(\theta)^2)}{\mathbb{E}[w(\theta, h) - 1]^2},$$

which is a generalization of the more widely-known Cramer-Rao bound and describes the minimal variance achievable by the estimator.

Under limited computational resources where full BP is infeasible, the only unbiased gradient estimation methods available are RLR, zeroth-order optimization, and RL. Compared to the latter two, RLR incorporates $h$-length half-order optimization and a single-step precise BP. According to Proposition 6.2 and the subsequent discussion, it is evident that RLR achieves the lowest variance among these methods.

## J.5 Proof of Theorem 6.4

*Proof.* Since $R(\cdot)$ is $L$-smooth, we have

$$
\begin{aligned}
E[R(\theta_{k+1})|\mathcal{F}_k] &= R(\theta_k) + \mathbb{E}\left[\langle \nabla R(\theta_k), \theta_{k+1} - \theta_k \rangle | \mathcal{F}_k\right] + \frac{L}{2}\mathbb{E}\left[\|\theta_{k+1} - \theta_k\|^2 | \mathcal{F}_k\right] \\
&= R(\theta_k) - \gamma\mathbb{E}\left[\langle \nabla R(\theta_k), \nabla R(\theta_k) + \epsilon \rangle | \mathcal{F}_k\right] + \frac{L\gamma^2}{2}\mathbb{E}\left[\|\nabla R(\theta_k) + \epsilon\|^2 | \mathcal{F}_k\right] \\
&\leq R(\theta_k) - \gamma(1 - \frac{L\gamma}{2})\|\nabla R(\theta_k)\|^2 + \frac{L\gamma^2\sigma_{\mathrm{RLR}}^2}{2} \\
&\leq R(\theta_k) - \frac{\gamma}{2}\|\nabla R(\theta_k)\|^2 + \frac{L\gamma^2\sigma_{\mathrm{RLR}}{}^2}{2},
\end{aligned}
\tag{17}
$$

where the last inequality holds if $\gamma \leq 1/L$. By taking expectations over the filtration $\mathcal{F}_k$, we have

$$
\mathbb{E}[R(\theta_{k+1})] \leq \mathbb{E}[R(\theta_k)] - \frac{\gamma}{2}\mathbb{E}[\nabla R(\theta_k)^2] + \frac{L\gamma^2\sigma_{\mathrm{RLR}}^2}{2},
$$

which is equivalent to

$$
\mathbb{E}[\nabla R(\theta_k)^2] \leq \frac{2}{\gamma}(\mathbb{E}(R(\theta_k)) - \mathbb{E}(R(\theta_{k+1}))) + \gamma L\sigma_{\mathrm{RLR}}^2.
$$

Taking the average over $k = 0, 1, \ldots, K$, we have

$$
\frac{1}{K+1}\sum_{k=0}^{K}[\mathbb{E}[\nabla R(\theta_k)^2]] \leq \frac{2(R(x_0) - R^*)}{\gamma(K+1)} + \gamma L\sigma_{\mathrm{RLR}}^2.
$$

Defining $\Delta_0 := R(x_0) - R^*$, if we set the step size

$$
\gamma = \left[\left(\frac{2\Delta_0}{(K+1)L\sigma_{\mathrm{RLR}}^2}\right)^{-\frac{1}{2}} + L\right]^{-1},
$$

then we have

$$
\frac{1}{K+1}\sum_{k=0}^{K}\mathbb{E}(\|\nabla R(\theta_k)\|^2) \leq \sqrt{\frac{8L\Delta_0\sigma_{\mathrm{RLR}}^2}{K+1}} + \frac{2L\Delta_0}{K+1},
$$

which completes the proof. $\qquad\square$

## J.6 Solution of Optimization Problem (7)

**Quadratic Upper Bound on Variance.** Using the variance decomposition bound (14), the objective of the problem (7) can be upper bounded by a quadratic function of $h$:

$$
Q(h) = a(h+1)^2 + b(h+1) + c, \tag{18}
$$

where the coefficients $a, b, c$ are given by:

$$
\begin{aligned}
a &= V_h^2 + V_z^2 - 2V_hV_z > 0, \\
b &= -2TV_z^2 + 2TV_hV_z,
\end{aligned}
$$

Then, the unconstrained optimal solution $h^\star$ is given by:

$$
h_{\mathrm{unc}}^\star = \frac{T(V_z^2 - V_hV_z)}{2(V_h^2 + V_z^2 - 2V_hV_z)} - 1. \tag{19}
$$

Where $h$ and $(T - 1 - h)$ are the number of steps for HO and ZO; $V_h^2$ and $V_z^2$ are coefficients indicating the magnitude of the variance of HO and ZO per step, and $V_h \ll V_z$ since HO has lower variance. Notice that $T > 2$. Therefore, $h_{\mathrm{unc}}^* > 0$. $\mathcal{B}_h$ and $\mathcal{B}_z$ are coefficients indicating the magnitude of the memory cost of HO and ZO per step and $\mathcal{B}_h > \mathcal{B}_z$. $\mathcal{B}_zT \leq \mathcal{B} \leq \mathcal{B}_hT$. The constraint of the problem (7) implies that $h \leq \frac{\mathcal{B}-\mathcal{B}_z(T-1)}{\mathcal{B}_h-\mathcal{B}_z}$, and $\frac{\mathcal{B}-\mathcal{B}_z(T-1)}{\mathcal{B}_h-\mathcal{B}_z} > 0$.

Then, the solution to the optimization problem is given by:

$$
\min\{\lfloor\frac{\mathcal{B} - \mathcal{B}_z(T-1)}{\mathcal{B}_h - \mathcal{B}_z}\rfloor, \lfloor\frac{T(V_z^2 - V_hV_z)}{2(V_h^2 + V_z^2 - 2V_hV_z)} - 1\rfloor\}.
$$

