# OpenReview forum: "Half-order Fine-Tuning for Diffusion Model: A Recursive Likelihood Ratio Optimizer"
_ICLR.cc/2026/Conference — ICLR 2026 Oral_

### Official Review · Reviewer_4kwL · 2025-10-29

**Soundness:** 3
**Presentation:** 3
**Contribution:** 3
**Rating:** 6
**Confidence:** 4

**Summary:**

This paper proposes an innovative "half-order" fine-tuning paradigm, making a substantial contribution to diffusion model optimization. By intelligently combining three gradient estimation strategies, it achieves a high level of theoretical and practical application. Rigorous mathematical proofs and extensive experimental validation demonstrate the RLR method's significant advantages over traditional approaches.

**Strengths:**

1. The innovative concept of "half-order" fine-tuning paradigm is proposed, which fills the gap between traditional first-order and zero-order methods.
2. FO, HO and ZO complement each other's strengths and find the optimal balance between variance and computational cost by optimizing the h and j parameters, taking into account the actual computational budget constraints.

**Weaknesses:**

1. The problem with FO is its high cost, and the problem with ZO is its high variance, but the author does not provide a clear analysis to explain this. For example, for a specific scenario, how many NFEs are needed for FO, ZO, and HO respectively, how is this calculated, what are the variances of these three, and why is there such a large variance problem. I think this needs a clearer analysis.
2. The visualization results look average, and the improvement is not significant enough. Lacks comparison with recent works such as FlowGRPO and ReFL.

**Questions:**

I wonder when I use ZO, when the noise is large, do I still need to go through the entire denoising trajectory to get the reward? If this is the case, then ZO also seems to have a high cost problem, which might be solved by process reward model such as SPO.

---

> ### Author Response · Authors · 2025-11-27
> **Rebuttal 1**
>
> > *[W1]. The problem with FO is its high cost, and the problem with ZO is its high variance, but the author does not provide a clear analysis to explain this.*
>
> We thank the reviewer for raising this question. In our implementation, FO, ZO, and HO all use the same number of network function evaluations (NFEs): each update step runs a single $T$-step diffusion chain, so the NFE per update is essentially $T$ for all three methods. The difference between FO, ZO, and HO is therefore not in NFE, but in **where the noise enters the gradient estimator** and how this affects variance and memory per step in the diffusion chain.
>
> FO backpropagates through the entire chain and must cache all intermediate latents to apply the chain rule, which yields the lowest-variance gradients but incurs the highest memory cost. ZO injects noise directly into the parameters (or uses a score-function estimator), which avoids caching latents but leads to the highest variance. **Our HO estimator instead leverages the inherent noise in the diffusion latents and performs FO-style backpropagation only on a short local sub-chain,** thereby explicitly trading off memory and variance under a fixed NFE budget.
>
> In this paper we chose not to re-derive the classical variance theory for FO and ZO, because this topic has already been extensively studied in the simulation and stochastic derivative estimation literature; see [1] and [2], which provides a systematic comparison of pathwise / IPA-type (FO) estimators and likelihood-ratio / score-function (ZO) estimators and formalizes the well-known fact that FO typically enjoys lower variance than ZO under standard regularity conditions. We verify the conditions proposed by [1] and [2] in Proposition 6.2 under the setting of diffusion model training to support our claims of the relationship of different estimator.
>
> Our contribution is instead to **introduce HO/RLR as a principled hybrid of FO and ZO along the diffusion chain under a fixed memory budget.** For this reason, our theoretical analysis focuses on (i) the step-wise variance decomposition of RLR in Appendix J.4, where we explicitly separate the FO-like and ZO-like contributions at different diffusion steps, and (ii) the constrained variance-minimization problem in Section 4.2, where, given a GPU memory budget, we derive how to choose the HO sub-chain length $h$ so as to optimally trade off FO-style low variance and ZO-style low memory.
>
> **We also provide empirical evidence in the paper to support our bias–variance discussions.** In Figure 3, we plot the convergence of truncated backpropagation with different numbers of BP steps. The results show that more BP steps lead to faster and better convergence, and also mitigate model collapse, consistent with the interpretation that longer BP reduces bias. In Figure 5, we compare the convergence behaviors of RLR, RL, and truncated BP (FO). Truncated BP converges quickly at the beginning but eventually suffers from model collapse; RL-based methods, which use unbiased but high-variance gradient estimators, converge much more slowly. In contrast, RLR converges rapidly while avoiding collapse, illustrating a favorable bias–variance trade-off. Furthermore, Table 9 examines performance under different HO sub-chain lengths $h$. Increasing $h$ consistently improves performance due to reduced variance, but at the cost of higher memory usage. Given the diminishing returns in performance gains as $h$ grows, alongside the steadily increasing memory overhead, we recommend $h=2$ as a practical default that achieves a strong balance between effectiveness and efficiency.
>
> [1] Cui, Zhenyu, et al. "On the variance of single-run unbiased stochastic derivative estimators." INFORMS Journal on Computing 32.2 (2020): 390-407.
>
> [2] Cui, Zhenyu, Yanchu Liu, and Ruodu Wang. "Variance comparison between infinitesimal perturbation analysis and likelihood ratio estimators to stochastic gradient." Operations Research Letters 50.2 (2022): 199-204.

---

> ### Author Response · Authors · 2025-11-27
> **Rebuttal 2**
>
> > *[W2]. The visualization results look average, and the improvement is not significant enough.*
>
> Thank you for pointing out recent related work such as FlowGRPO and ReFL. We agree these methods are closely related in spirit, and we will clarify our relationship to them more explicitly.
>
> Conceptually, FlowGRPO is a specific RL-style preference optimization method. A key difference, however, is that **FlowGRPO explicitly relies on multiple generations per prompt** (sampling several trajectories for the same prompt at each update), whereas our main RL baseline (DDPO) and our RLR variants use one sample per prompt per update under a fixed NFE budget. In other words, a faithful reproduction of FlowGRPO’s training scheme would require substantially **more function evaluations (NFEs) per optimization step** than the baselines we compare against. This makes a direct, “apples-to-apples” comparison under the same compute / NFE budget non-trivial and, in our opinion, not strictly fair. We will make this point explicit and position FlowGRPO as a higher-compute RL variant that is orthogonal to our work.
>
> Methodologically, ReFL is essentially a truncated backpropagation (FO) method for diffusion alignment: it reduces bias by backpropagating through a limited number of forward steps. In the AlignProp paper, **AlignProp has already been shown to outperform ReFL** on the same tasks. Since we include AlignProp as a strong truncated-BP baseline in our experiments, a separate comparison to ReFL would be largely redundant: if RLR is competitive with or superior to AlignProp, it is implicitly competitive with ReFL as well. We will clarify this baseline selection rationale and add explicit references to both ReFL and FlowGRPO in the related-work discussion.

---

### Official Review · Reviewer_KciP · 2025-11-01

**Soundness:** 3
**Presentation:** 3
**Contribution:** 3
**Rating:** 8
**Confidence:** 3

**Summary:**

This paper addresses the crucial task of efficiently aligning diffusion models (DMs) to meet downstream application requirements after pre-training.   Contemporary fine-tuning methods, such as Reinforcement Learning (RL) and truncated Backpropagation (Truncated BP), suffer from high variance and biased gradient estimation, respectively.  The authors propose the Recursive Likelihood Ratio (RLR) optimizer, a novel "Half-Order" (HO) fine-tuning paradigm. RLR constructs a new gradient estimator by cleverly combining First-Order (FO), Half-Order (HO), and Zeroth-Order (ZO) estimators within the recursive chain. The paper theoretically proves that the RLR estimator is unbiased (overcoming the defect of truncated BP) and has a lower variance than RL/ZO methods. Extensive experiments on text-to-image and text-to-video tasks validate the superiority of RLR. It not only outperforms baseline methods (like DDPO and Alignprop) on multiple reward benchmarks, but critically, it also avoids the "model collapse" problem caused by truncated BP9. Furthermore, the paper proposes a novel prompt technique called "Diffusive Chain-of-Thought" (DCoT), which synergizes naturally with RLR's HO estimator, allowing the model to optimize for specific generation scales (e.g., "fine-grained" details).

**Strengths:**

- This paper addresses the crucial task of efficiently aligning foundation diffusion models. This represents a highly important and practically valuable problem.
- This paper proposes the Recursive Likelihood Ratio (RLR) optimizer, a novel "Half-Order" (HO) fine-tuning paradigm that successfully overcomes these challenges. T
- he paper also introduces a novel prompt technique that synergizes naturally with the RLR optimizer, further enhancing the originality of the contribution.
- The paper rigorously demonstrates its method's advantages in terms of bias, variance, and convergence from both theoretical and experimental standpoints, while also proving its practical effectiveness.

**Weaknesses:**

- The paper exhibits significant inconsistencies in its core methodology description, particularly regarding the sampling strategy for the Half-Order (HO) sub-chain starting point, $j$. In Section 4.2 (Methodology), the paper describes $j$ as being sampled from a categorical distribution based on gradient norms. However, in Section 5.3 (DCoT Experiment), $j$ is described as being selected from a uniform distribution ($j \sim \mathcal{U}(1, T-h)$). This contradictory description makes it impossible to determine which sampling strategy the standard implementation of RLR is supposed to use.
- Furthermore, the implementation of DCoT (Diffusive Chain-of-Thought) introduces a critical external dependency. As shown in Section 5.3 and Appendix F, DCoT relies on an external Large Language Model (LLM) to generate the 'coarse-mid-fine' grained prompts. This raises doubts about its robustness in practical deployment.

**Questions:**

- As in weakness, why different descriptions occur, and how to choose the sampling strategy?

---

> ### Author Response · Authors · 2025-11-27
> **Rebuttal**
>
> > *[W1]. The paper exhibits significant inconsistencies in its core methodology description*
>
> Thank you for carefully checking the description of how we sample the starting point $j$ of the Half-Order (HO) sub-chain. You are right that Sections 4.2 and 5.3 currently use different wording, which is confusing.
>
> Our standard implementation of RLR (used in all Text2Image and Text2Video experiments in Sections 5.1 and 5.2) always samples $j$ from the gradient-norm–based categorical distribution described in Section 4.2, i.e. $j\sim\mathcal{J}(1,T-h)$, where $\mathcal{J}$ is constructed from $\text{softmax}(\Vert g_1 \Vert,\dots,\Vert g_{T-h} \Vert)$.
>
> The sentence in Section 5.3 that says "the HO estimator term uniformly picks a starting point $j\sim U(1,T-h)$" is an unfortunate typo. In the DCoT experiments, we do not change the core RLR sampling strategy; instead, **we restrict the HO sub-chain to those diffusion steps corresponding to the scale we want to enhance** (e.g., the hand structure), as the text after that sentence tries to explain: we constrain $j$ to lie in a window $[a,b]$(we set $a=30$ and $b=40$) of "deficient" steps and then apply HO only there ($j\sim\mathcal{J}(a, b)$, where $\mathcal{J}$ is constructed from $\text{softmax}(\Vert g_a \Vert,\dots,\Vert g_{b} \Vert)$).
>
> > *[W2]. Furthermore, the implementation of DCoT (Diffusive Chain-of-Thought) introduces a critical external dependency.*
>
> Thank you for pointing this out. We would like to clarify the role of DCoT in our paper. The DCoT (Diffusive Chain-of-Thought) setup is not part of the core RLR algorithm, but an **auxiliary diagnostic experiment** designed to illustrate that RLR **can capture and enhance visual information at different semantic scales (coarse–mid–fine).**
>
> The standard RLR method is straightforward to implement and has no dependency on any external LLM. The HO component only requires the final scalar reward value and the log-probability of the latent in the diffusion chain, which can be implemented in practice as `(R(x_0).detach() * latent.log_prob()).backward()`. The length of the HO sub-chain is controlled simply by detaching the computation graph at the terminal step of the HO sub-chain. This implementation is lightweight, model-agnostic, and can be easily integrated into a wide range of diffusion models.
>
> Indeed, DCoT uses an external LLM only to generate the coarse/mid/fine prompt variants needed for this specific multi-scale analysis. We agree that this external dependency may limit the practicality of DCoT as a generic deployment recipe, and we will clarify in the revision that (i) DCoT is an optional analysis tool rather than a mandatory component of RLR, and (ii) in practice, the same kind of multi-scale prompts could also be obtained from simple hand-crafted templates or domain-specific heuristics if desired.
>
> We will update Section 5.3 and Appendix F to clearly separate the core, LLM-free RLR method from the DCoT diagnostic protocol and explicitly acknowledge the external-LLM dependency of DCoT as a limitation specific to that experiment, not to RLR itself.

---

### Official Review · Reviewer_joeE · 2025-11-01

**Soundness:** 3
**Presentation:** 3
**Contribution:** 3
**Rating:** 6
**Confidence:** 3

**Summary:**

The paper proposes a Half-order Fine-tuning method for efficiently adapting large-scale diffusion models (e.g., Stable Diffusion) to downstream datasets.  The authors also present a theoretical proof that RLR is unbiased, has lower variance, and enjoys convergence guarantees. The experiments demonstrate RLR’s efficiency.

**Strengths:**

- The paper presents a novel fine-tuning scheme and devises gradient estimators for the diffusion model’s chain-of-thought, which appears genuinely innovative.
- The theoretical analysis is careful and offers a credible justification for the proposed approach.

**Weaknesses:**

- Missing a comparison with related diffusion model fine-tuning baselines, such as D3PO[1].
- The experiments are limited to SD 1.4 and SD 2.0, which are now dated. Moreover, the method’s generalization to the Flux architecture remains unclear.

$\text{[1] Yang, Kai, et al. "Using human feedback to fine-tune diffusion models without any reward model." Proceedings of the IEEE/CVF Conference on Computer Vision and Pattern Recognition. 2024.}$

**Questions:**

NA.

---

> ### Author Response · Authors · 2025-11-27
> **Rebuttal**
>
> > *[W1]. Missing a comparison with related diffusion model fine-tuning baselines, such as D3PO.*
>
> Thank you for the suggestion regarding additional baselines. Originating from optimizing a Bradley–Terry–style reward model with a KL constraint with reference policy, D3PO [2] directly applies a direct preference optimization objective to train the diffusion model using human-annotated preference pairs. We also include another baseline, DPOK [1], which combines KL regularization with a policy-gradient update.
>
> DPOK and DDPO are both policy gradient-based methods. The key difference is that DPOK studies KL regularization when training a separate DM for each prompt, whereas DDPO trains on many prompts simultaneously using clipping and importance sampling. In our setting, our method also trains on many prompts at once. Therefore, DPOK can be seen as a variant of DDPO with KL regularization. D3PO utilizes the DPO loss to train the DM. Given preference pairs, the DM can be updated directly. The results, shown in the following table using the Pick-a-Pic prompt suite, demonstrate that our RLR method consistently outperforms both baselines.
>
> | Method | PickScore | HPSv2 | AES | ImageReward |
> |-|-|-|-|-|
> | DPOK | 17.66 | 22.98 | 5.63 | 50.25 |
> | D3PO | 17.93 | 25.47 | 6.02 | 63.76 |
> | RLR | 20.14 | 28.57 | 6.53 | 75.65 |
>
> [1] Fan, Ying, et al. "Dpok: Reinforcement learning for fine-tuning text-to-image diffusion models." Advances in Neural Information Processing Systems 36 (2023): 79858-79885.
>
> [2] Yang, Kai, et al. "Using human feedback to fine-tune diffusion models without any reward model." Proceedings of the IEEE/CVF Conference on Computer Vision and Pattern Recognition. 2024.
>
> > *[W2]. The experiments are limited to SD 1.4 and SD 2.0, which are now dated. Moreover, the method’s generalization to the Flux architecture remains unclear.*
>
> We agree that SD-1.4 and SD-2.0 are no longer the most recent diffusion backbones, and we will explicitly acknowledge this as a limitation in the revised version. Our primary goal in this work, however, is not to chase absolute SOTA quality, but to study the variance–memory trade-offs of RL-based and BP-based alignment under a fixed compute budget and insure unbiasedness. For this purpose, SD-1.4/2.0 remain standard and widely used benchmarks in the alignment literature, and they allow us to compare directly to strong existing baselines such as DDPO, D3PO, and AlignProp on exactly the same backbones.
>
> Within this setting and our limited computational resource, **we deliberately designed a diverse evaluation suite to test the robustness of RLR:** we include both SD-1.4 and SD-2.0 backbones, both text-to-image and text-to-video tasks（finetune on videocrafter）, and multiple reward models. Across these heterogeneous setups, RLR consistently improves sample quality and alignment under the same NFE / GPU-memory budget, which supports our main claim that a properly designed HO hybrid can effectively trade off bias and variance in large-scale generative models.
>
> Regarding Flux-style architectures: modern flow-matching / rectified-flow models (e.g., Flux) are **still implemented as multi-step recursive inference through a shared backbone**. As such, they face essentially the same structural issue as diffusion models: gradients must propagate through many solver steps, leading to **a similar tension between first-order backprop (low variance, high memory) and RL/ZO-style estimators (unbiased but high variance).** RLR is defined exactly at this level—as a gradient estimator for a multi-step generative chain under a memory budget—and does not rely on any SD-specific architectural assumption. Conceptually, the same HO framework can be applied to Flux by defining the HO sub-chain over the flow integration steps.
>
> Practically, however, adapting and re-training a full Flux-scale model with RL-based preference optimization would require substantial engineering changes to our current codebase and significantly more computational resources than we currently have access to. We view it as important future work. We will clarify this in the paper and emphasize that our contributions are architecture-agnostic at the level of the training objective and gradient estimator, even though our current experiments focus on SD-style backbones.

---

### Author Response · Authors · 2025-11-30
**General Responses**

We thank the AC and reviewers for their thorough feedback and reviews. Our paper introduces RLR, a Half-Order (HO) gradient estimator for preference-based alignment of diffusion models under a fixed GPU-memory / NFE budget. Conceptually, RLR is a intelligent hybrid of first-order (FO) backprop and zero-order (ZO) / RL-style estimators along the diffusion chain: it runs a single $T$-step chain (same NFE as FO/ZO/RL), but only backpropagates through a short local sub-chain and uses a ZO estimator elsewhere, explicitly trading off variance vs. memory.

On the empirical side, we now make our baselines clearer and stronger. In addition to DDPO and AlignProp, we explicitly position and compare against D3PO and DPOK: D3PO applies a DPO-style objective directly on diffusion models using preference pairs, while DPOK combines KL regularization with policy gradient. Across SD-1.4/2.0 and our text-to-image / text-to-video setups, RLR consistently outperforms these RL/DPO-style baselines under the same NFE and memory constraints. Since AlignProp has already been shown to outperform ReFL, using AlignProp as our truncated-BP baseline makes a separate ReFL comparison largely redundant.

Our theoretical contribution is specific to our novel HO estimator: in Appendix J.4 we provide a step-wise variance decomposition of RLR along diffusion steps, and in Section 4.2 we formulate a memory-constrained variance-minimization problem that yields an optimal HO sub-chain length $h$. These analyses are supported by convergence plots (FO, RL, RLR) and ablations over $h$, showing how increasing $h$ reduces variance but increases memory, with $h=2$ providing a good effectiveness–efficiency trade-off.

We also clarify two methodological points. First, in all main experiments the HO starting step is sampled from a gradient-norm–based categorical distribution; the “uniform” wording in the DCoT section is a typo. DCoT merely restricts the range of eligible steps (coarse/mid/fine) and then applies the same sampling rule. Second, DCoT is an auxiliary diagnostic protocol, not part of RLR itself: it uses an external LLM only to construct coarse/mid/fine prompts, whereas the core RLR algorithm is LLM-free and simple to implement. We believe the same HO/RLR framework is conceptually applicable to modern flow-matching architectures such as Flux, and we view extending our study to these newer models as promising future work.

---

### Meta-Review · Area_Chair_awbt · 2025-12-07

**Summary:**

The reviewers' initial concerns primarily focused on three areas:

## Baselines and Comparisons

Reviewers noted a lack of comparison with relevant recent methods such as D3PO, FlowGRPO, and ReFL to fully establish the method's superiority.

## Experimental Settings
There were concerns that the experiments relied on older diffusion backbones (Stable Diffusion 1.4 and 2.0) rather than newer architectures like Flux, and that the visual improvements were not significant enough.

## Methodological Clarity
One reviewer identified a contradiction in the text regarding the sampling strategy for the Half-Order sub-chain (described as categorical in one section and uniform in another). There was also concern that the "Diffusive Chain-of-Thought" (DCoT) component created an impractical dependency on external LLMs.

**Reviewer Concerns:**

Addressed by the rebuttal:

Missing Baselines (joeE, 4kwL): The authors successfully addressed this by providing new experimental results comparing their method (RLR) against D3PO and DPOK, showing RLR outperformed them. They also logically justified excluding ReFL (as AlignProp is a stronger version of the same concept and was already compared) and FlowGRPO (which requires a higher compute budget).

Inconsistency in Sampling Strategy (KciP): The authors clarified that the mention of "uniform distribution" was a typo. The standard implementation uses the gradient-norm-based categorical distribution.

External LLM Dependency (KciP): The authors clarified that DCoT is an optional diagnostic tool, not a core part of the RLR algorithm, which is itself LLM-free.

Gradient Estimator Analysis (4kwL): The authors provided a clear explanation that FO, ZO, and HO are compared under a fixed NFE budget and pointed to the theoretical variance decomposition provided in the appendix.

Still Outstanding:

Reliance on older models (joeE): While the authors argued that SD 1.4/2.0 are standard for studying alignment trade-offs, the limitation of not testing on state-of-the-art models like Flux remains an empirical gap, even if the theoretical justification is sound.

Visual Quality (4kwL): The reviewer felt the visual improvements were "average". While the authors pointed to quantitative metrics (PickScore, ImageReward) to refute this, subjective visual quality assessments often remain a point of contention.

**Reviewer Scores:**

Reviewer joeE (Current: 6): Likely increase. The reviewer's main weakness was the missing comparison to D3PO. The authors provided this data, showing RLR's superiority. The concern about older SD models remains but was defended well.

Reviewer KciP (Current: 8): Likely maintain 8. This reviewer was already positive. The rebuttal resolved the confusion regarding the text inconsistency (the typo) and the LLM dependency. The reviewer would likely feel more confident in their high score.

Reviewer 4kwL (Current: 6): Likely maintain 6. The authors addressed the theoretical questions about variance and NFE costs effectively. However, if the reviewer remains unconvinced by the visual results, the score might not jump significantly.

---

### Decision · Program_Chairs · 2026-01-26

Accept (Oral)